# Ocean heat forced West Antarctic Ice Sheet retreat after the Last Glacial Maximum

Elaine M. Mawbey[1,6], James A. Smith [1,6] ✉, Claus-Dieter Hillenbrand [1], Katharine R. Hendry [1], Erin L. McClymont [2], Mervyn J. Greaves [3], Johann P. Klages [4], Gerhard Kuhn [4], Svetlana Radionovskaya [1], Charlotte L. Spencer-Jones[2], Robert D. Larter [1], Julia S. Wellner [5] & Pierre Dutrieux [1]

The West Antarctic Ice Sheet (WAIS) is thinning at an accelerating rate, driven by melting at its margins by warm Circumpolar Deep Water (CDW). However, this understanding is largely based on observations from recent decades, leaving the long-term influence of ocean temperature on WAIS stability uncertain. Here we reconstruct bottom water temperatures and water mass properties over the past 18 kyr using benthic foraminiferal Mg/Ca and $\delta^{13}C$ records from sediment cores in the Amundsen Sea. Our data indicate that warm CDW occupied the continental shelf between ~18.0 and 10.1 kyr BP, coincident with major WAIS retreat from the shelf break to near its present-day grounding-line position along the Marie Byrd Land coast. Bottom waters cooled after ~10.1 kyr BP and remained relatively stable thereafter, with no evidence for substantial grounding-line migration. Continued atmospheric warming across West Antarctica until a mid-Holocene thermal maximum (~6−3 kyr BP) without further retreat indicates that ocean heat was the primary driver of WAIS variability since the Last Glacial Maximum.

The net loss of glaciers flowing into the Amundsen Sea Embayment (ASE), which accounts for ~35% of the total discharge from the WAIS, has increased by 77% since 1973[1,2]. Ice sheet thinning has been accompanied by rapid inland retreat of the grounding lines of Pine Island Glacier and Thwaites Glacier potentially initiating an unstable and irreversible retreat[1,3,4]. Complete collapse of the glaciers in this region would raise global sea level by ~1.5 m, substantially above the 'likely' modelled range of 0.25–0.95 m predicted for the 21st century by the IPCC[5]. The coherent thinning of glaciers across the ASE has been attributed to basal melting of the floating ice shelves by warm CDW upwelling onto the continental shelf[6] (Fig. 1). Originating from the Southern Ocean, modified CDW (CDW hereafter) is channelled along landward deepening bathymetric troughs, eroded by palaeo-ice

streams[6] (Fig. 1). At the ice sheet margin, CDW is ~3.8 °C above the in-situ freezing point and causes vigorous basal melting of the fringing ice shelves that buttress the inland ice[7].

Oceanographic measurements show the volume and temperature of CDW in Pine Island Bay increased between 1994 and 2009[7], and this increase coincided with progressive thinning of the ice shelf[8], glacier speed-up and enhanced ice discharge into the ocean. Ocean models[9] attribute these incursions of CDW to changes in the frequency and strength of the westerly winds near the continental shelf break. Typically these winds are driven by convective anomalies in the equatorial Pacific troposphere (El Niño) that radiate an atmospheric wave train toward the region[10], strengthening winds at the continental shelf edge and leading to greater upwelling of CDW onto the shelf. Wind-induced

[1]British Antarctic Survey, High Cross, Madingley Road, Cambridge, UK. [2]Department of Geography, Durham University, Lower Mountjoy, South Road, Durham, UK. [3]The Godwin Laboratory for Palaeoclimate Research, Department of Earth Sciences, University of Cambridge, Downing Street, Cambridge, UK. [4]Alfred-Wegener-Institut Helmholtz-Zentrum für Polar- und Meeresforschung, Am Alten Hafen 26, Bremerhaven, Germany. [5]Department of Earth and Atmospheric Sciences, University of Houston, Houston, TX, USA. [6]These authors contributed equally: Elaine M. Mawbey, James A. Smith. ✉e-mail: jaas@bas.ac.uk

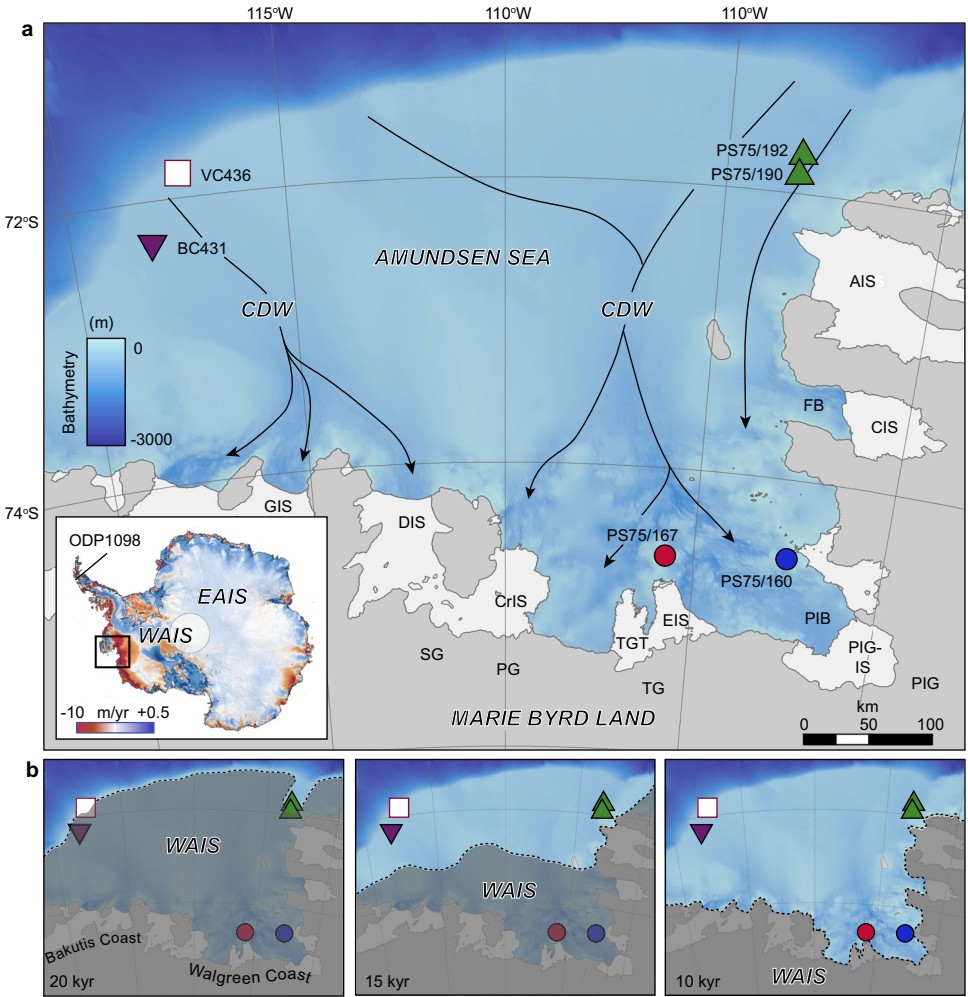

**Fig. 1 | I Maps of the Amundsen Sea Embayment and reconstructed glacial history. a** Bathymetric map with core locations (coloured symbols) showing the main pathways of warm Circumpolar Deep Water (CDW; black line). Grey shaded areas: grounded ice; white areas: ice shelves. Bathymetry is derived from BedMachine[88] https://nsidc.org/data/nsidc-0756/versions/3. Coastlines are from the SCAR Antarctic Digital Database (accessed 2021). https://www.scar.org/resources/antarctic-digital-database/ available under a CCBY 4.0 license (https://creativecommons.org/licenses/by/4.0/deed.en) following the information at this link (https://scar.org/library-data/maps/add-digital-database). No changes were made. **b** Reconstructed glacial history 20–10 kyr[22–25] showing extent of grounded ice (dark shading) relative to core locations. Note that the grounding line (dotted line) had retreated close to its modern position by ~10 kyr. Inset (left corner panel

a), shows the location of the study area within the wider context of Antarctica, with colours showing mass loss from Antarctica (2003–2019)[89] available under a CCBY 3.0 license (https://creativecommons.org/licenses/by/3.0/deed.en) following the information at this link (https://digital.lib.washington.edu/researchworks/items/b60f8b8c-ecd4-4f3c-a0be-5c9201dbd894/full). No changes were made. Highest mass loss rates (red shading) occur along the Pacific-facing coast of West Antarctica, driven by warm CDW. WAIS: West Antarctic Ice Sheet; EAIS: East Antarctic Ice Sheet; PIG: Pine Island Glacier; PIG-IS: Pine Island Glacier Ice Shelf; PIB: Pine Island Bay; TG: Thwaites Glacier; TGT: Thwaites Glacier Tongue; EIS: Eastern Ice Shelf; PG: Pope Glacier; SG: Smith Glacier; AIS: Abbot Ice Shelf; CIS: Cosgrove Ice Shelf; CrIS: Crosson Ice Shelf; DIS: Dotson Ice Shelf; GIS: Getz Ice Shelf; Ferrero Bay: FB. Also shown are the Walgreen and Bakutis coasts (panel **b**).

Ekman pumping may also influence the depth of the on-shelf thermocline and thus drive higher melt rates[8,11,12]. One model[13] was able to relate two recent periods of accelerated thinning of Pine Island Glacier in 1974−87 and after 1994 to simulated increases in CDW influx. Conversely, another model[8] was able to show—in combination with ocean measurements—that from 2010 to 2012 oceanic melting of the Pine Island Glacier ice shelf decreased by 50% in response to a major La Niña event. The weak convection in the tropical Pacific resulted in decreased wind stress at the shelf edge, reduced upwelling of CDW and reduced glacier melting in Pine Island Bay, thereby providing a mechanism by which current mass loss could be reduced. Together these studies demonstrated that the mean-ocean state in the Amundsen Sea can vary—at least on annual to decadal time-scales[14]—and provides a mechanism that might increase the rate of ice sheet melting and retreat, or in the case of 2012, reduce localised ice sheet thinning[15].

However, with only a few decades of ocean temperature data available for the ASE, it remains difficult to assess whether CDW upwelling onto the shelf has been relatively constant, driven on annual to decadal timescales by tropical forcing, or has varied substantially in the past, potentially even promoting thickening and advance of grounded ice during times of reduced upwelling. Atmospheric and ocean modelling studies attempted to address this. Simulations revealed that the mean easterlies were progressively weakened by anthropogenic radiative forcing throughout the twentieth century[16], causing tropically forced decadal warm ocean anomalies to become more prevalent at the ice-sheet margin[17]. Episodes of particularly strong wind anomalies in the mid-1940[18,19], drove strong melting that caused ice streams to unground from seabed ridges[20] and progressively retreat due to ice-ocean feedbacks and increased anthropogenic forcing. However, these studies only examined the last century of change and did not consider long term changes in warm CDW

incursions onto the Antarctic shelf and their link with periods of increased glacier discharge. Hillenbrand et al.[21] used a multi-proxy approach to reconstruct water mass variations in the ASE throughout the Holocene (11.7 kyr BP to present). Using the stable carbon isotope ($\delta^{13}C$) composition of benthic foraminifera *Trifarina angulosa* and preliminary Mg/Ca data (n = 10), they argued that enhanced CDW upwelling could have forced deglaciation of the innermost part of Pine Island Bay from at least -10.4 kyr BP until -7.5 kyr BP. But at the time of publication, no suitable Mg/Ca temperature calibration existed for *T. angulosa*, meaning that the long-term relationship between ice sheet retreat and ocean heat content remained uncertain.

Given the modelled sensitivity of the WAIS to CDW-induced melting on decadal[14] and millennial timescales[21] and the potential uncertainty in the magnitude of future global sea-level rise from the ASE drainage sector, there is a fundamental need to constrain past oceanographic changes—and particularly ocean temperature—in order to understand whether the ocean-driven melting we see today is unique, has operated in the past (and if it has driven ice sheet retreat), or, if retreat is reversible.

Here we present Mg/Ca and $\delta^{13}C$ data of benthic foraminiferal calcite (*T. angulosa*) for 6 sites in the ASE (Fig. 1; Supplementary Data 1, 3), which together provide a unique record of water mass variability for the past -18 kyr. No single sedimentary record from the ASE contains sufficient *T. angulosa* shells throughout this entire period because sites were either covered by grounded ice or affected by iceberg turbation at some point since the Last Glacial Maximum (LGM) (see Fig. 1b), so we used a multi-site approach. Furthermore, calcareous microfossils, required not only for analysing the geochemical composition of their shells but also for establishing reliable radiocarbon-based age models, are absent in the vast majority of previously collected cores[22–25]. Indeed, the cores used here offer the best carbonate preservation and are the only ones suitable and available for this type of work. An estimate of BWT is calculated using an Antarctic-wide calibration for *T. angulosa*[26], which demonstrated the sensitivity of this species to BWT based on core-top samples from the Amundsen Sea, Antarctic Peninsula, Weddell Sea and East Antarctic margin (Methods) (Supplementary Fig. 1). *T. angulosa* was used for three reasons: (1) it is common in modern and post-LGM sediments on the Antarctic continental shelf, and particularly in the Amundsen Sea; (2) it is genetically related to *Uvigerina* spp., which has previously been used for BWT reconstructions[27]; and (3) it is a shallow infaunal species, which is considered less likely to be influenced by the effects of carbonate ion concentration on the Mg/Ca ratios of their tests than epifaunal species[27] (see Methods). To aid interpretation of the Mg/Ca record and provide independent constraints on water mass variability, we also measured the $\delta^{13}C$ composition of *T. angulosa* as a proxy for the carbon isotope composition of the dissolved inorganic carbon ($\delta^{13}C_{DIC}$) of the ambient seawater. Because the dissolved inorganic carbon of the two dominant water masses in the Amundsen Sea (CDW and Antarctic Surface Water (AASW)) are distinct, $\delta^{13}C$ measurements of foraminiferal calcite can be used as a reliable tracer for its source water mass variations[21]. Age constraints for the six cores are based on previously published age data[23–25,28] that are supplemented by additional AMS $^{14}C$ dates on calcareous microfossils (VC436) and $^{210}Pb$ data (BC431) (Supplementary Fig. 4,5; Supplementary Data 3, 4).

## Results and discussion
### Mg/Ca and $\delta^{13}C$ records of ocean temperature and water mass variability
Resulting BWT, calculated from measured Mg/Ca ratios, are -4.4 to +4.2 °C (Supplementary Data 1) and are unrealistic given the freezing point of regional surface waters (-1.87 °C) and maximum CDW temperature (+2.0 °C) in the Southern Ocean[29]. Temperatures outside of a realistic range likely reflect limitations of our calibration (see Methods). Because of this, we do not interpret the Mg/Ca-ratios in terms of

absolute temperature change(s) and instead focus on trends between 'warmer' and 'cooler' BWT. While this does not allow direct comparisons between *palaeo* and measured (modern) temperatures, it does enable us to investigate the response of the ice sheet, which is well established[22–25,30], to changes in ocean forcing. Furthermore, regardless of which calibration or method for calculating the temperature data is used, all curves display consistent trends (Supplementary Fig. 1a, b and Fig. 7 in ref. [26]), suggesting that although estimates of BWT may improve as calibrations evolve, the observed trends are robust. This assumption is supported by the $\delta^{13}C$ data whereby high BWT correspond to low $\delta^{13}C$ values that have previously been shown to be indicative of CDW[21]. Thus, and consistent with previous work[29], we attribute high Mg/Ca and low $\delta^{13}C$ ratios to enhanced CDW delivery/ warm conditions whereas low Mg/Ca and high $\delta^{13}C$ values indicate a dominance of AASW-like/cooler conditions.

The Mg/Ca and $\delta^{13}C$ data are presented in Fig. 2, split between outer and inner shelf sites (Fig. 1). Outer shelf variations are largely constrained by data from site VC436, which was the first to deglaciate following the LGM[25] and spans the post-LGM to Early Holocene period (-18.0 to 9.8 kyr BP). The Holocene part of VC436, after 9.8 kyr BP, is heavily disturbed due to iceberg turbation[25], making this interval unsuitable for Mg/Ca and isotope analyses. Relative to inner shelf sites, VC436 is characterised by high Mg/Ca and low $\delta^{13}C$ values implying that warm CDW-like conditions persisted throughout this interval. Highest Mg/Ca values occur around -11 kyr BP while lowest values are recorded 14.5–13.9 kyr BP. Data from PS75/190 and PS75/192 in the northeastern ASE, are consistent with warmer BWT during the Late Glacial but are restricted to a single datapoint in the post-LGM to Early Holocene period ( - 18.0 to 9.8 kyr BP). Mg/Ca and $\delta^{13}C$ data from PS75/190 and PS75/192 additionally suggests a transition to cooler BWT and overall reduction in CDW during the Holocene (Fig. 2), although the precise timing of this change is poorly constrained due to the limited age constraints. BC431, which only recovered sediment spanning the past 150 years, indicates that Late Holocene BWTs were cooler than from -18.0 to 9.8 kyr BP.

Inner shelf sites PS75/160-1 and PS75/167 (Fig. 2b) span the time interval after -10.46 kyr BP, when ice had retreated to a position close to present[22]. They are characterised by a shift to lower Mg/Ca and higher $\delta^{13}C$ ratios that are indicative of cooler BWTs and an overall reduction of CDW after -10.0 kyr BP. In detail, Mg/Ca is characterised by periods of lower values 8.9–7.1 kyr BP, 5.2–4.8 kyr BP and 1.8–1.1 kyr BP that are coeval with higher $\delta^{13}C$ ratios and periods of slightly higher Mg/Ca and lower $\delta^{13}C$ ratios at 6.3–5.6 and -3.6 kyr BP, respectively. Generally, both outer and inner shelf records display centennial to millennial-scale water mass variability during the past 18 kyr but note that the resolution of our dataset is insufficient to resolve seasonal variability.

While our composite dataset has limited temporal overlap (-660-year) between outer and inner shelf sites we note two important observations: (1) there is a gradient from high Mg/Ca and low $\delta^{13}C$ ratios to low Mg/Ca and high $\delta^{13}C$ ratios during the Early Holocene (-10–8.0 kyr BP) (Fig. 2a–d). This gradient is better resolved in the $\delta^{13}C$ data (Fig. 2a, b) because fewer specimens are required for isotopic analyses, thus allowing measurement of more sediment core intervals and increasing temporal resolution. Furthermore, sites PS75/190 and PS75/192, while also limited in their temporal resolution, provide Mg/ Ca and $\delta^{13}C$ data spanning the past -16 kyr BP (Fig. 2a, c). The Mg/Ca and $\delta^{13}C$ data at these sites are characterised by cooler BWT after -10 kyr BP compared to the period 18–10 kyr BP, thereby supporting our interpretation of an embayment-wide change in water mass properties during the earliest Holocene. (2) There is a strong argument that outer shelf oceanographic conditions are representative of the entire shelf because present-day oceanographic observations around Antarctica indicate consistency between the inner and the outer shelf[31]. Presently, and irrespective of the proximity to the ice sheet grounding line, BWTs on the Antarctic shelf are either warm or cold, with little variation

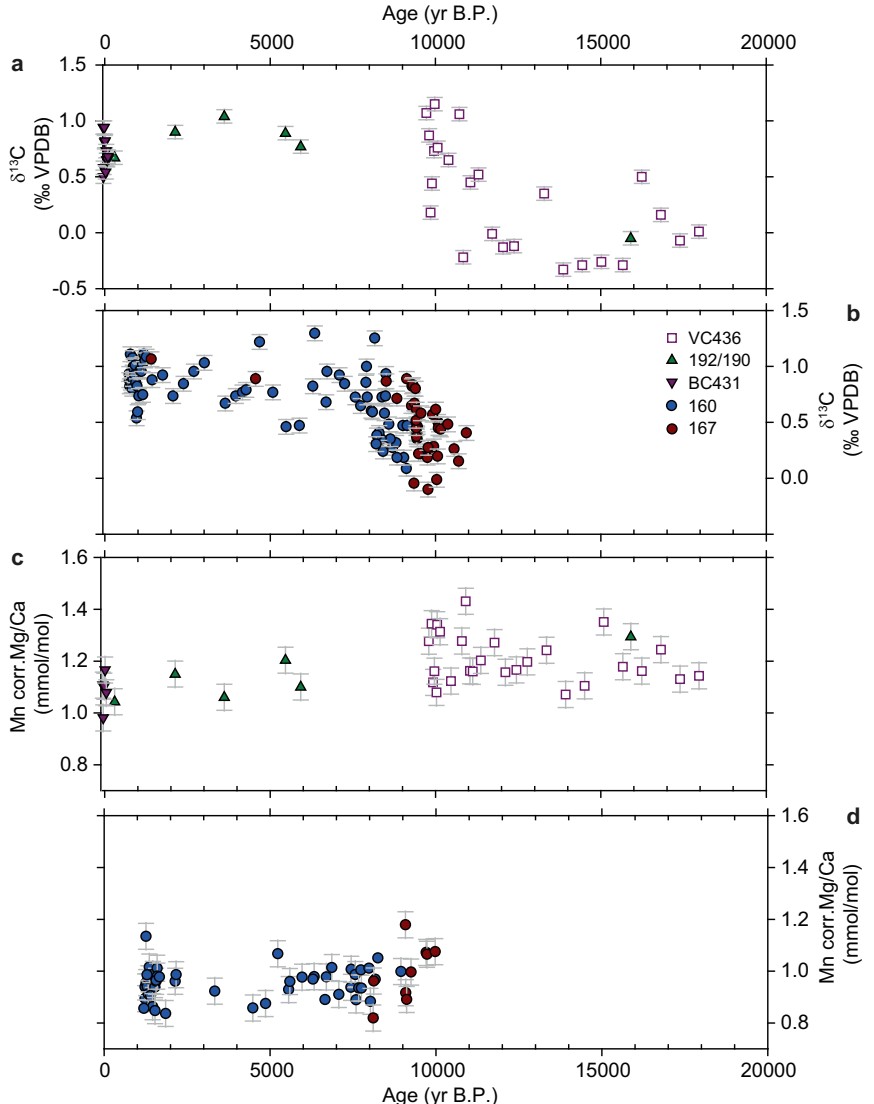

**Fig. 2 | I Outer and Inner shelf Mg/Ca and δ¹³C data from the Amundsen Sea cores. a** Outer shelf δ¹³C values of benthic foraminifera *T. angulosa* from cores VC436, PS75/190/192 and BC431. **b** Inner shelf δ¹³C values of benthic foraminifera *T. angulosa* from cores PS75/160 and PS75/167 from Hillenbrand et al.[21]. **c** Outer shelf Mg/Ca ratios of benthic foraminifera *T. angulosa* corrected for diagenetic coating of shells using the Mn/Ca ratio (see Methods and Supplementary Fig. 7) from cores VC436, PS75/190/192 and BC431. **d** Inner shelf Mn corrected Mg/Ca ratios of benthic foraminifera *T. angulosa* from cores PS75/160 and PS75/167. Error bars represent 2 standard deviations (s.d.) (see Methods).

between the ice front and the shelf break[31]. Furthermore, numerical modelling demonstrates that the residence time of shelf waters, i.e., the time required to fully replace water properties through advection following a regime shift, is just a few years for small shelf seas such as the Amundsen Sea[32] and up to a few decades for large ones such as the Weddell Sea[33]. Thus, a signal observed in one deep location that is coherent over centuries, is likely to be representative of an entire shelf sea even if the ice sheet configuration changes. Accordingly, our composite dataset indicating warmer BWT on the outer shelf ~18.0 to 10 kyr BP and cooler BWT on the inner shelf after ~10-8.0 kyr BP applies equally to all shelf areas free of grounded ice. Change-point analysis of the composite Mg/Ca and δ¹³C datasets (see Methods) confirms that the transition from warm to cooler BWT occurred between 10.1 and 7.9 kyr BP, with peak BWT identified between 11.0 and 10.1 kyr BP (Methods and Supplementary Fig. 6).

### Timing of ice sheet and water mass variations since the LGM

Extensive marine geological data from the Amundsen Sea shelf indicate that during the LGM, the WAIS advanced to a position at, or

close to the continental shelf break[23,25,34−36] (Fig. 1b). Grounding line retreat was underway by ~22-19 kyr BP, reaching the mid-shelf at ~13.8–13.5 kyr BP and a position close to modern along the Walgreen, Bakutis and Hobbs Coast (hereafter referred to as Marie Byrd Land coast) before ~11.2 kyr BP, without significant change since then[22] (Fig. 1b). This time-slice reconstruction is supported by ice sheet modelling, which simulates the majority of Amundsen Sea mass loss as having occurred between 17 and 9 kyr BP[37]. Surface exposure dating in the hinterland around Pine Island Bay documents a period of rapid, but localised, ice sheet thinning between 9 and 6 kyr BP, which has been linked to the collapse of a more extensive ice shelf covering the inner shelf at 7.5 kyr BP, resulting in debuttressing and drawdown of inland ice[21,38]. Several other studies also point to localised glacier and ice shelf changes during the past 10 kyr, such as the Early-to-Mid-Holocene retreat of the Cosgrove Ice Shelf[39] and Late Holocene ice-sheet thinning in the vicinity of Pope and Thwaites glaciers[40] but without significant increases in mass loss. Sea-level data[40] support the scenario of Early Holocene deglaciation followed by relatively stable ice positions until recent times and suggest that

Thwaites and Pine Island glaciers—which are currently the main contributors of WAIS mass loss—have not been substantially smaller than present during the Holocene. Our reconstruction of BWT and $\delta^{13}C$ data indicates that warmer, CDW-like conditions were persistent on the outer shelf throughout the deglacial phase (-18–10.1 kyr BP) (Figs. 1b and 3f). Conversely, cooler BWT and a shift to higher $\delta^{13}C$ values indicative of more AASW-like deep water properties are recorded after -10.1–7.9 kyr BP, when WAIS grounding lines along the Marie Byrd Land coast are considered to have been largely stable[22,24]. This provides compelling evidence that post-LGM WAIS deglaciation was primarily driven by incursion of warm CDW. We now consider the role, and possible interplay between other climatological drivers, focusing first on the deglaciation.

## CDW as the primary driver of ice-sheet retreat following the LGM

The long-held view is that deglaciation of the WAIS following the LGM was driven by a combination of rising sea levels—primarily due Northern Hemisphere deglaciation—warming ocean waters, and changes in atmospheric circulation and/or ice sheet dynamics[41]. Our combined Mg/Ca and $\delta^{13}C$ dataset provides direct evidence that WAIS retreat in the Amundsen Sea was linked to warm water forcing, whereby high BWTs are recorded on the outer shelf as early as -18 kyr BP, concomitant with initial retreat of the WAIS from its expanded LGM position (Fig. 1b). Today, incursions of CDW onto the ASE shelf are tightly coupled to enhanced wind anomalies at the shelf edge, driven by variability in the Amundsen Sea Low[42], and modulated by both El

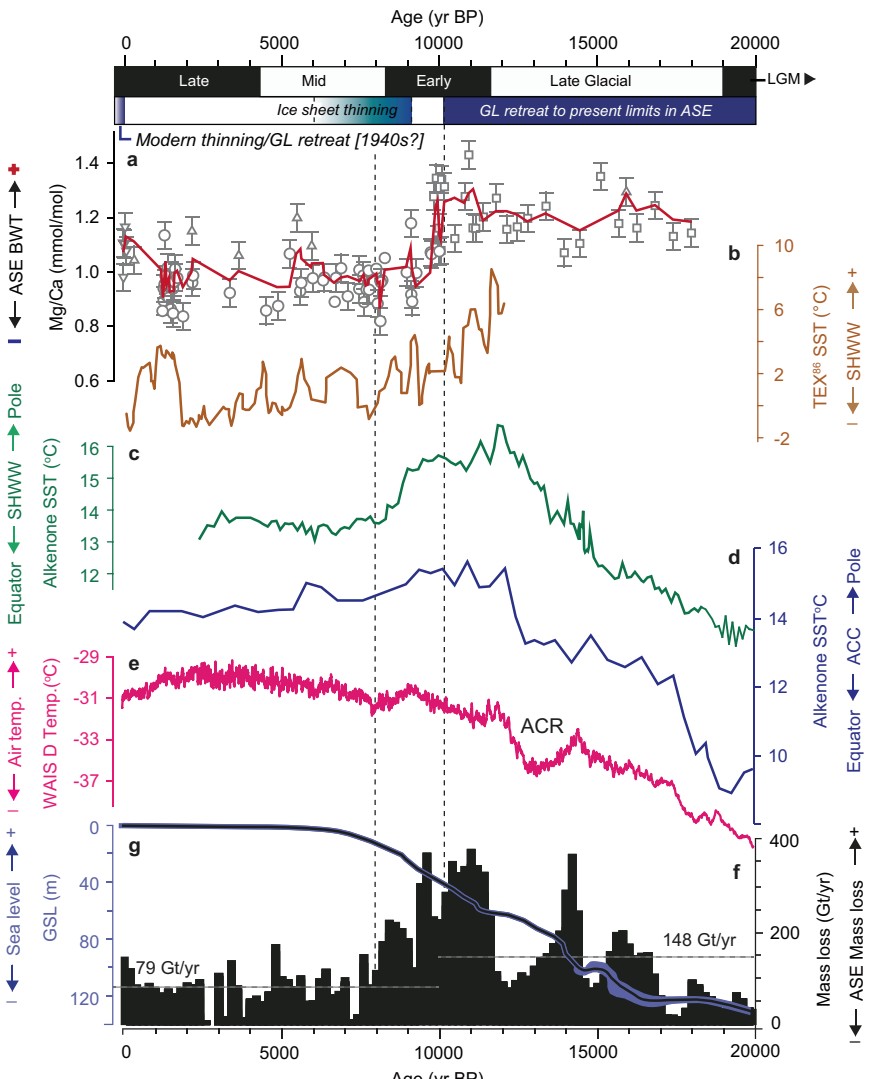

**Fig. 3 | I Comparison of palaeoceanographic changes in the Amundsen Sea Embayment (ASE) with glacial history (top axis; GL: grounding line) and (sub-) Antarctic climate proxy records. a** Mn corrected Mg/Ca data (3-point moving average; red line) for the Amundsen Sea shelf, indicating warmer/cooler bottom water temperature (BWT) throughout the past -18 kyr. The vertical dashed line denotes significant change-points 10.1–7.9 kyr BP in the Mg/Ca (and $\delta^{13}C$) data. Error bars represent 2 standard deviations (s.d.) (see Methods). **b** TEX$_{86}$ sea surface temperatures (SST) from Palmer Deep (ODP Site 1098; Fig. 1a), western Antarctic Peninsula shelf[50], showing a similar ocean (SST) cooling during the Late Glacial-Early Holocene. The SST data at this site capture an integrated (0–200 m) sub-surface record of ocean temperature, which has been interpreted in terms of changes in the strength of the Southern Hemisphere westerly winds (SHWW) **c** Alkenone SST records from South Pacific cores MD-97120 southwest of New Zealand[48] and **d** ODP Site 1233 west of Chile[52], as a proxy for the position of the SHWW. Increasing (decreasing) temperature indicates poleward (equatorward) SHWW movement. **e** Surface air temperatures from the WAIS Divide ice core[90], reaching a Mid-Holocene thermal maximum -6-3 kyr BP. Antarctic Cold Reversal (ACR) occurs at 14.7–13.0 kyr BP, corresponding to a minor shift to cooler BWT on the Amundsen Sea shelf. **f** Mass loss from the Amundsen Sea Embayment (ASE) drainage sector of the WAIS inferred from ice-sheet modelling experiments[37], binned into 200-year intervals. Horizontal grey dashed lines denote average mass loss from 18 to 10 kyr BP and 10-0 kyr BP, respectively. A - 47% reduction in mass loss is observed for the period 10-0 kyr BP relative to 18-10 kyr BP. **g** Reconstructed global sea-level (GSL) with 95% probability limiting values (blue shading)[91].

Niño events[8,19,43] and the phase of the Southern Annular Mode (SAM)[44,45]. During La Niña, a cooler ocean state is characterized by a deepened thermocline that separates cold surface water from warmer CDW, reducing the contact area between CDW and the ice-shelf base[8]. Ultimately, a deeper thermocline reduces the volume of warm water in contact with the ice-shelf base[46], resulting in reduced basal melting, slower ice shelf thinning and a slower decline of glacier buttressing and consequently, less ice discharge[8]. Furthermore, Hillenbrand et al.[21] linked the Early Holocene (~10.4 to 7.5 kyr BP) transition from warm CDW-like to cooler Antarctic Surface Water (AASW)-like conditions in the Amundsen Sea to a northward (equatorward) shift of the SHWW. This interpretation was based on $\delta^{13}C$ gradients (low to high) between cores PS75/167 and PS75/160, and supported by multi-proxy data from Campbell Island[47] and the southwest Pacific Ocean[48], which together suggest a pole-to-equator migration of the westerlies after 10 kyr BP. A weakening of the SHWW is also inferred from decreasing sea surface temperature (SST) data on the western Antarctic Peninsula shelf after ~9 kyr BP[49,50]. Our findings support this hypothesis and additionally reveal a pronounced decrease in BWT after ~10.1 kyr BP, coinciding with the hydrographic transition from CDW- to AASW-like conditions (Fig. 2b).

These results suggest that the mean latitudinal position of the SHWW played a dominant role in modulating CDW inflow onto the shelf during the Early Holocene but its role in on-shelf incursions of CDW immediately after the LGM remains uncertain. Southern Ocean planktic foraminiferal $\delta^{18}O$ records indicate a ~4° equatorward shift and about a 25% weakening of the SHWW during the LGM compared to the Mid-Holocene (about 6,000 years ago)[51] while alkenone-derived SST reconstructions from Chatham Rise[48] (east of New Zealand) and the east coast of Chile[52] record major warming between ~19.2 and ~12.1 kyr BP (Fig. 3c). The warming trend is linked to the poleward displacement of the SHWW[47] and the Antarctic Circumpolar Current (ACC)[52]. A ~6° poleward shift of the ACC following the LGM is further supported by biomarker-based SST reconstructions from the southern Indian Ocean[53]. The latitudinal position of the SHWW thus emerges as a key control on the exchange of warm water onto the Amundsen Sea shelf. A poleward-shifted SHWW would bring the ACC closer to the continental margin, promoting upwelling and enhancing basal melting. A similar mechanism has been proposed for deglaciation in the western Ross Sea (JOIDES Trough)[54] where retreat after ~19.6 kyr BP coincides with higher subsurface ocean temperatures linked to a southward SHWW shift. However, a notable feature of our CDW record is that's it is largely binary through the deglaciation rather than displaying a gradual increase in BWT as the SHWW and ACC migrated poleward. This possibly suggests the existence of a latitudinal threshold beyond which CDW incursions are triggered, or that other mechanisms were also influential. In this context, ocean-driven ice sheet retreat following the LGM may have been amplified by rising sea levels (Fig. 3e, g), with additional positive feedback between sea-level rise and the landward deepening Amundsen Sea shelf[25]. Indeed, Smith et al.[23,25] previously linked a speed-up in grounding line retreat rates in the Amundsen Sea around 14 kyr BP to a positive feedback between sea-level rise (and particularly meltwater pulse 1 A) and the retrograde seabed, which deepens significantly inboard of the mid-shelf. In contrast, surface air temperatures likely played a subordinate role to CDW and sea-level during deglaciation, given their limited impact on the current mass balance in the Amundsen Sea sector. Past atmospheric warming would have needed to exceed modern levels by several degrees to drive substantial retreat via surface melting, which is not supported by ice core data[55]. The ultimate cause of latitudinal shifts of the SHWW is debated but likely involved large-scale climate teleconnections[56]. Menviel et al.[57] proposed that a cessation of North Atlantic Deep Water (NADW) formation during Heinrich Stadial 1 (HS1: 18.2–14.6 kyr BP) cooled the North Atlantic and warmed the South Atlantic, shifting the Inter-Tropical Convergence Zone southward

which strengthened and shifted the SHWW poleward. In the Amundsen Sea, our data indicate that this reorganisation led to enhanced upwelling of warm CDW onto the shelf, which in turn drove WAIS retreat. However, a fully coupled ocean-atmosphere-ice sheet modelling is needed to fully elucidate the links between global climate, the SHWW position and the interplay between CDW, rising sea level and ice-sheet retreat in the Amundsen Sea.

## Divergent drivers of ice sheet change during the Holocene?

Consistent with the earlier study of Hillenbrand et al.[21] we link the well-documented grounding line stability along the Marie Byrd Land coast during the Early Holocene[22,30] to the observed reduction in CDW upwelling after ~10.1–7.9 kyr BP. An additional observation is that surface air temperatures continued to increase for several millennia, reaching a Mid-Holocene optimum (Fig. 3e), yet had no significant impact on the grounding lines of Pope, Smith, Thwaites and Pine Island glaciers. This is consistent with the idea that CDW was the dominant driver of past ice-sheet variations and supports modelling results, which indicate that warm water accessing ice shelf cavities and the ice-sheet margin leads to faster retreat[58]. Conversely, once the ocean driver is removed or muted, the grounding line can stabilise on high points in the bed[15]. However, while the grounding lines of Pope, Smith, Thwaites, and Pine Island glaciers remained relatively stable during the Holocene, ice shelves and glaciers on the eastern side of the embayment continued to evolve[39,59]. The Early-to-Mid-Holocene retreat of the Cosgrove Ice Shelf has been primarily attributed to atmospheric warming, and specifically the strengthening of the Amundsen Sea Low[39] linked to warming in the tropical Pacific. This may imply divergent drivers during the Holocene, whereby variations in WAIS retreat along the Marie Byrd Land coast were muted in response to a reduction in CDW, while other parts of the glacier system remained sensitive to enhanced or peak atmospheric warmth from ~6 to 2 kyr BP. In this context, the depth of the thermocline, modulated by shelf-edge wind stress or other thermodynamic atmospheric processes, might be the critical factor governing past ice-sheet variability. Modelling indicates two 'threshold' thermocline depths (700 and 1000 m)[46]. When the thermocline shoals above 700 m, grounding lines becomes highly sensitive to melt forcing, potentially triggering rapid and irreversible ice-sheet retreat[46]. Conversely, when the thermocline is lowered below 1000 m, melt rates may be sufficiently reduced to allow ice-shelf thickening and theoretically, ice-sheet re-grounding. A similar threshold may have operated in the past, whereby oceanic conditions in the Amundsen Sea oscillated between warm and cool states on centennial to millennial timescales depending on thermocline depth. During the Holocene, a more northerly position of the SHWW/ACC reduced the influx of warm CDW onto the shelf, deepening the thermocline—similar to conditions observed during contemporary La Niña phases[8]. Under these conditions, incursions of warm CDW onto the shelf still occurred, but at greater depth, below our core records from above 800 m water depth, and therefore did not induce significant ice-shelf or ice-sheet melting along the Marie Byrd Land coast. A deeper thermocline may also help reconcile microfossil data suggesting that incursions of CDW persisted in deep basins, such as Ferrero Bay, during the Mid-to-Late Holocene[59]. However, the benthic foraminifer *Bulimina aculeata*, used in Minzoni et al. as an indicator of CDW, may not be uniquely diagnostic, because it is only present in some modern surface sediments presently bathed in CDW[60]. Finally, the role of the SHWW in modulating mid-to-late Holocene oceanographic conditions appears complex. While multiple lines of evidence point to an equatorward SHWW shift during the Early Holocene coincident with reduced BWT and muted ice-mass variations in the Amundsen Sea sector[47], other data suggest that the latitudinal position of the SHWW varied throughout the Holocene without consistently affecting BWT in the Amundsen Sea[61,62]. Again, this could indicate additional drivers are important i.e., growing influence of ENSO during the Holocene[63], or

simply reflect the variable temporal resolution of our Holocene palaeoceanographic dataset, which is unlikely to capture higher-frequency, decadal-scale variability. What is clear however, is that the amplitude of ocean thermal forcing was greatest during the post-LGM deglaciation (18-10.1 kyr BP) and was reduced during the Holocene, perhaps intensifying again in the twentieth century in response to a combination of consistent ENSO during the 1940s followed by enhanced anthropogenic warming after the 1950s[43].

## Implications for future WAIS retreat

Glaciers in the Amundsen Sea drainage sector are retreating rapidly, with no indication that these changes will be reversed in the coming century[4]. Contemporary mass loss is linked to the poleward transport of warm Circumpolar Deep Water (CDW), which has intensified in recent decades[8,9,17], with the magnitude of ocean-driven melting being considered the key variable determining future sea-level rise from Antarctica[64]. Our reconstruction of BWT and water-mass variability ($\delta^{13}C$) suggests that similar processes operated in the past, whereby embayment-wide ice-sheet retreat coincided with the presence of warmer CDW−likely tied to the poleward displacement of the SHWW/ACC following the LGM. Conversely, when the SHWW/ACC were displaced equatorward, reduced CDW inflow limited the delivery of warm water onto the shelf, allowing glacier advance (as during the LGM) or, in the Holocene context, to a period of WAIS grounding-line stability along the Marie Byrd Land coastline. Given that climate models predict a consistent strengthening and poleward displacement of the SHWW in the future[65,66], due to enhancement of the SAM[67,68] as well as a trend toward stronger and more frequent El Niño[69], the WAIS will likely continue to retreat through mechanisms similar to those reconstructed for the past. Furthermore, since the subglacial beds of Pine Island and Thwaites glaciers deepen inland, pulses of rapid grounding line retreat that have been documented in the past[70] may become more frequent in the future, amplified by positive feedbacks associated with widening ice-shelf cavity geometries and marine ice sheet instability[25,71].

## Methods

### Core material and chronology

Our coring strategy targeted shallower sites because prior experience indicated that this increases the likelihood of recovering records that contain calcareous microfossils[72] but are situated deep enough to capture a CDW signal. Gravity cores PS75/192 (71.74′S, 103.32′W) and PS75/190 (71.87′S, 103.38′W) were collected from the outer shelf (Abbot Trough) during expedition ANT-XXVI/3 of RV Polarstern in 2010, with core recoveries of 2.14 and 2.96 m, respectively[28]. Water depth at site PS75/192 was 793 m and 775 m at site PS75/190. The age models for these sites are from Klages et al.[28] and were generated by AMS $^{14}C$ dating on calcareous microfossils in combination with the measurement of relative paleomagnetic intensity (RPI). All previously published ages are presented as calibrated years before present (where present is 1950 CE), unless otherwise stated. New $^{14}C$ dates from core VC436 generated for this study were converted to calendar ages using the Marine20 radiocarbon calibration[73] applying a delta ΔR of 813 ± 30 years (#522 in the marine radiocarbon database, http://calib.org/marine/). Calibration was performed using Calib v. 8.1.0. Abundance of calcareous foraminifera is variable downcore but yielded sufficient foraminifera to measure stable isotopes ($n = 7$) and trace elements ($n = 5$) in a handful of horizons. Gravity cores PS75/160 (74.56′S, 102.62′W) and PS75/167 (74.62′S, 105.80′W) were collected on the inner shelf of Pine Island Bay during expedition ANT-XXVI/3[21]. Core recovery at PS75/160 (337 m) and PS75/167 (526 m) were 6.7 m and 9.3 m, respectively. The chronology of these cores has been constrained using AMS $^{14}C$ dating on calcareous microfossils[21]. Core PS75/160 spans an age range from 9.2 kyr BP to 1.2 kyr BP. Absence of sediments younger than -1.2 kyr BP implies the loss of seafloor surface

sediment during coring or its recent erosion by either iceberg scouring or intense bottom currents. Core PS75/167 spans an age range from 10.4 kyr BP to 8.2 kyr BP. Lack of calcareous microfossils in the uppermost sediments likely reflects site-specific factors, such as local sea-ice cover or calcite dissolution.

Core BC431 (72.30′S, 118.16′W) was collected using a box corer (BC) on expedition JR141 with RRS James Clark Ross in 2006 from a water depth of 512 m. The chronology of BC431 is based on a constant rate of supply (CRS) $^{210}Pb$ age model[74] (Supplementary Fig. 4,5). $^{210}Pb$ and $^{137}Cs$ activities were measured on 1 cm thick sediment slices by gamma-spectrometry using Canberra ultralow-background Ge-detectors in the Department of Geography, Durham University. The core showed surface contents of unsupported $^{210}Pb$ of about 81 mBq/g with a tendency for exponential decline with depth in the upper 6.5 cm (Supplementary Fig. 4). Below about 6 cm core depth the $^{210}Pb$ activity was at the detection limit or lower. $^{137}Cs$ concentration was below the detection limit throughout. The 5.9 m long vibrocore VC436 (71.81′S, 117.43′W) was also recovered on expedition JR141 from a water depth of 479 m. The chronology of VC436 is from Smith et al.[25] and newly generated AMS $^{14}C$ ages on calcareous microfossils. Chronological data indicate an age reversal at -60 cm depth in this core, which has been attributed to nearby iceberg scouring[25]. However, here we focus only on the interval from 580 to 160 cm, which yielded $^{14}C$ dates in stratigraphic order with an age range of 17.97−9.79 kyr BP.

### Trace metal analyses and BWT calibration

Sediment samples were washed through a 63 μm sieve and dried for 24 h at 40 °C at the Godwin Laboratory for Palaeoclimate Research at the Department of Earth Sciences, University of Cambridge (UK). The >63 μm fraction was then dry-sieved and specimens of T. angulosa were picked from the 250−355 μm fraction. All picked specimens of T. angulosa belonged to the costate (ribbed) morphotype[26]. T. angulosa is a shallow infaunal species living 0−5 cm below the seafloor surface which is considered less likely to be influenced by the effects of carbonate ion, $[CO_3^{2-}]$, concentration on the Mg/Ca ratios of their tests than epifaunal species[27]. In this environment, porewaters tend to equilibrate rapidly with respect to carbonate chemistry, and the saturation state remains near zero. This minimizes the influence of ambient $[CO_3^{2-}]$ on calcification[27,75]. Elderfield et al.[27] demonstrated this effect for Uvigerina spp. and given the close genetic relationship between Uvigerina and Trifarina spp.[26,76], it is reasonable to assume that Trifarina responds similarly−i.e., that any $[CO_3^{2-}]$ effect on Mg/Ca is negligible.

Samples were visually inspected for preservation state, and no obvious signs of degradation/alteration were observed (Supplementary Fig. 9). Approximately 30 specimens were gently crushed between two glass plates and visible contaminants were removed using a fine paintbrush. The fragments then underwent a cleaning procedure adapted from Boyle and Keigwin[77]. This includes: (i) the removal of clay by ultrasonication of test fragments 3x in 18.2 MΩ·cm deionised water and 3x in methanol; (ii) a reductive step to remove any metal oxides using a solution of hydrous hydrazine and citric acid in ammonia; (iii) an oxidative step using a solution of hydrogen peroxide in sodium hydroxide to remove organic matter; and (iv) a dilute nitric acid leach using Optima pure nitric acid. Between the mechanical clay removal step and the reductive step, samples were examined under a binocular microscope and any obvious non-carbonate particles removed. The cleaned samples were dissolved in ultra-pure nitric acid (0.1 M $HNO_3$), centrifuged, and a 'concentrate' aliquot was transferred to a new sample vial to prevent leaching from any remaining contaminants. Trace metal data was analysed using ICP-OES following the method of de Villiers et al.[78]. Long-term instrumental precision of element ratio data determined by replicate analyses of a consistency standard prepared at University of Cambridge with a magnesium/calcium ratio = 1.3 mmol/mol is ±0.55%. Accuracy of Mg/Ca determinations was

confirmed by replicate analyses of standard JCt-1, where mean Mg/Ca of 1.264 ± 0.005 mmol/mol (r.s.d. 0.43%), measured on 13 occasions during this study, was obtained, in agreement with published results[79]. The cleaning efficiency was monitored by measuring Mn/Ca and Fe/Ca[80]. Maximum Mn/Ca and Fe/Ca are 1.06 and 0.19 mmol/mol (ave. 0.25 and 0.05 mmol/mol) respectively. A weak positive correlation is observed between Mn/Ca and Mg/Ca for the different records (Supplementary Fig. 8). To assess the impact of any diagenetic Mg contribution on the Mg/Ca data we applied a correction to the Mg/Ca data, assuming a Mg/Mn diagenetic coating of 0.15 ± 0.05 mmol/mol following Hillenbrand et al.[21]. This value reflects the average composition of Mn nodules and ferromanganese crusts from the South Pacific and the Pacific sector of the Southern Ocean[81,82]. The corrected Mg/Ca values exhibit the same overall down-core trend as the uncorrected data, though with reduced amplitude, with elevated Mg/Ca ratios prior to -10 kyr BP (Supplementary Fig. 8a–e). Importantly, the use of alternative Mg/Mn ratios (e.g., 0.20 and 0.25 mmol/mol)[82] for diagenetic coatings does not alter the observed difference between the deglacial and Holocene periods (Supplementary Fig. 8a-e), supporting the robustness of this transition. Propagating both sources of analytical error ($\sigma_{Mg/Ca} = 0.005$ mmol/mol and $\sigma_{Mg/Mn} = 0.05$ mmol/mol) yields a mean uncertainty of ± 0.0503 mmol/mol for the Mn corrected Mg/Ca dataset. This was further explored using a Monte Carlo approach to probabilistically evaluate the uncertainties (Supplementary Fig. 1a). Monte Carlo simulations ($n = 10000$) produce a mean 2 s.d. uncertainty of ± 0.0577 mmol/mol (Supplementary Data 1).

Calculation of BWT followed the Antarctic-wide calibration for *T. angulosa* from Mawbey et al.[26], which used surface sediment samples from various sectors of the Antarctic shelf. As noted above, applying this calibration (green line Supplementary Fig. 2a) produces an unrealistic temperature range (Supplementary Data 1). Overestimation of ocean temperatures is unlikely to be an analytical error, given our close approximation of Mg/Ca in reference materials/standards. We also rule out diagenetic or reworking issues because of the excellent preservation state of individual tests, which were all inspected under light microscope prior to analysis. Rather, the unrealistic BWT likely reflects the increased uncertainty in the calibration for low BWT due to the limited number of data points in this range. It would be possible to apply a scaling factor of 0.3 and remain within the confines of the 95% confidence intervals of the calibration (blue line Supplementary Fig. 2) to produce BWT closer to the measured modern-day temperature range (-1.89 to +1.79 °C; Supplementary Data 1) but this would require verification by further modern surface sediment calibration. This approximation might offer an alternative input for ice-sheet modelling studies[83,84], which aim to investigate the role of ocean forcing on ice-sheet variability. Such studies have previously used outputs from an archived atmospheric-oceanic global climate model simulation[85] that potentially omits regional variability. Propagation of the replicate error ($\sigma_{Mg/Ca} = 0.005$ mmol/mol), diagenetic coating error ($\sigma_{Mg/Mn} = 0.05$ mmol/mol) and the calibration error ($\sigma_{calib} = 0.098$ mmol/mol) gives a 2 s.d. uncertainty of ± 1.6 °C for calculated BWT (Supplementary Fig. 2). The calibration uncertainty is estimated following Cléroux et al.[86], based on the standard error of the regression coefficient reported by Mawbey et al.[29].

## Stable carbon isotope analyses

In total 3-5 specimens of *T. angulosa* were hand-picked under a binocular microscope from the >250 μm fraction. $\delta^{13}C$ analyses were undertaken using a IsoPrime dual inlet mass spectrometer with Multiprep device at the NERC Isotope Geosciences Laboratory in Keyworth (UK). The $\delta^{13}C$ data are reported as per mil (‰) deviations of isotopic ratios ($^{13}C/^{12}C$), calculated to the VPDB scale using a within-run laboratory standard calibrated against NBS-19 standards. Analytical reproducibility of an in-house calcite standard (KCM) is <0.1‰ for the $\delta^{13}C$ data. The analyses had an average 2 s.d. uncertainty of 0.06‰ on a

2‰ standard for $\delta^{13}C$ across the analytical period. The $\delta^{13}C$ of the samples were corrected by +0.85‰ following Hillenbrand et al.[21] to account for species specific vital effects.

## Change point analysis

Standard mean change point (CP) analysis was calculated using the 'changepoint' package in R[87] to determine the periods when significant trend changes occurred in the Mg/Ca and $\delta^{13}C$ datasets (Supplementary Fig. 6). Analysis was performed on both the split (inner and outer shelf) and composite record. Segmented mean analysis of the inner shelf data revealed change points (CPs) at 8248 yr BP for Mg/Ca and 7977 yr BP for $\delta^{13}C$. For the outer shelf record, the Mg/Ca dataset exhibited two CPs, at 11043 yr BP (indicating a shift to warmer BWT) and 10050 yr BP (shift to cooler BWT). A single CP at 10911 yr BP was identified for the outer shelf $\delta^{13}C$ data. Analysis of the composite record yielded consistent CPs at 10135 and 9113 yr BP for Mg/Ca, and 10911 and 7938 yr BP for $\delta^{13}C$, which is used to define the transition from warmer (10135 yr BP) to cooler (7938 yr BP) BWTs. We attribute the minor offsets between Mg/Ca- and $\delta^{13}C$-derived CPs to the higher resolution of the $\delta^{13}C$ record.

## Data availability

Trace metal (Mg/Ca, Mn/Ca), $\delta^{13}C$ and chronological (radiocarbon, $^{210}Pb$) data are included as Supplementary Data 1–4 and is also available in the Polar Data Centre database https://www.bas.ac.uk/data/uk-pdc/.

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

## Acknowledgements

This study was funded by the Natural Environment Research Council (NERC) grant NE/M013081/1, awarded to J.A.S., E.L.M., K.R.H., and C.D.H. Additional funding was provided by the THOR project, a component of the International Thwaites Glacier Collaboration (ITGC) via National Science Foundation (NSF) grant 173894 and NERC grant NE/S006664/1. We thank the captains, crews, shipboard scientists, and support staff participating in RRS *James Clark Ross* expeditions JR141 and RV *Polarstern* expedition ANT-XXVI/3 (PS75). This is ITGC Contribution No. ITGC-162.

## Author contributions

J.A.S. conceived the idea for the study and secured funding through NERC grant NE/M013081/1 with C.D.H., E.L.M., and K.R.H. J.A.S. and E.M.M. wrote the initial draft of the manuscript with input from C.D.H., K.R.H., E.L.M., and S.R. E.M.M. analysed the trace metals with MG and measured stable isotopes on the foraminifer shells. S.R. helped interpret the trace metal data and performed the Monte Carlo error simulations. P.D. provided oceanographic data and insight regarding contemporary ocean variability. G.K., C.D.H., J.A.S., and J.P.K. collected the PS75 cores while C.D.H., J.A.S., and R.D.L. collected the JR141 cores. J.A.S., C.D.H.,

J.P.K. developed the age-models for the JR141 and ANT-XXVI/3 cores. C.S.J. performed the $^{210}$Pb measurements on core BC431 with input from J.S.W. S.R., M.G., R.D.L., J.P.K., G.K., P.D., C.S.J., and J.S.W. also commented on the manuscript and provided input to its final version.

## Competing interests

The authors declare that they have no known competing interests or personal relationships that could have appeared to influence the work reported in this paper.
