## [Transparent Peer Review file · Nature Communications]

Ocean heat forced West Antarctic Ice Sheet retreat after the Last Glacial Maximum

Corresponding Author: Dr James Smith

Version 0:

Reviewer comments:

Reviewer #1

(Remarks to the Author)

Review for NCOMMS-25-17907-T

Mawbey et al. present carbon isotope and Mg/Ca data for the Amundsen Sea Embayment (ASE), a critical region for understanding the trajectory of future sea-level rise and relate this to the presence/absence of Circumpolar Deep Water (CDW), a probable driver of ice loss. The study extends records beyond the Holocene and is therefore an important contribution to understanding the long-term nature of the West Antarctic Ice Sheet. However, I find several problems that need to be addressed before publication:

1. D13C and Mg/Ca records from all sites are combined in Fig. 2 and 3a to represent changes in CDW versus AASW for the ASE in the text. However, Site PS75/167 and PS75/160 is in the inner ASE, several hundred kilometres from PS75/191, PS75/190, VC436 and BC431 on the outer shelf. These records should be separated and discussed separately: one record for the outer shelf discussing CDW upwelling; the other for the inner ASE discussing incursion further south (although this is already covered in Hillenbrand et al. 2017 but the Mg/Ca can be used to confirm these findings).
2. Following from the previous point, I think the records of outer shelf CDW presence/absence should be highlighted as the main finding of this study as these are new records that extend to ~18 ka BP. Any conclusions for the inner ASE should be toned down as these stem from the previous study of Hillenbrand et al. (2017) and start at ~10 ka BP.
3. The trends in the data need to be handled statistically instead of relying on the reader to pick out the perceived trends in the data, which are subjective. For example, change-point analysis could be used to determine if the jump at ~10 ka is statistically significant. I expect the d13C would see a significant jump, but the Mg/Ca ratios would not. This should be done after the inner ASE records (PS75/167 and PS75/160) are separated from the outer ASE (PS75/191, PS75/190, VC436 and BC431).
4. You mention the carbonate ion eject but don't state how you determined if this eject was influencing your data. Some evidence to show constant carbonate ion concentrations or some other screening method would be vital here. The ASE is under influence from large amounts of freshwater discharge from nearby glaciers suggesting variable alkalinity across the shelf. Carbonate preservation also appears to be highly variable suggesting a dynamic carbonate ion concentration across the shelf.

5. The wider interpretation is very selective and focuses on a marine geoscience perspective, a common problem among marine paleoclimate records, whilst neglecting to mention other drivers of ice loss since the LGM such as atmospheric and sea level forcing. More southerly SHW may drive CDW incursions onto the shelf, but it also brings warm, moist air to West Antarctica leading to surface melting. Sea-level rise from the LGM to Holocene would have destabilised ice sheets leading to dynamic calving. It is likely the combination of warm ocean and atmospheric currents and sea level rise would have combined to drive ice loss post LGM. This paper supports the ocean forcing contribution which is important but not the whole story. A wider context is therefore needed.

6. We see a gradual increase in sea level and southward transition of the SHW in Fig. 3, but Mg/Ca ratios are flat. Why do we not see gradually more CDW on the shelf with this change? This suggests the $\delta^{13}\text{C}$ and Mg/Ca ratios are instead displaying a binary signal, i.e., CDW is either present or not, which needs to be discussed. Furthermore, other records for SHW migration show a continued poleward migration until the mid-Holocene. This may suggest ocean forcing is important before 10 ka but atmospheric or sea level forcing is important after 10 ka – an interesting finding that has not been discussed.

After these changes are made, I expect you will have an important study of outer shelf CDW changes that can inform on the contribution of ocean forcing on ASE ice loss, and maybe more muted CDW influence on continued loss during the Holocene. This finding should be placed within the context of other important forcing mechanisms such as atmospheric currents and sea-level rise.

Other comments:

Fig. 1 b and text referring to ASE reconstructions: this reconstruction and the relevant text refers to the pre-Holocene records and might be misleading. Several studies since have shown extensive ice shelf retreat during the Holocene. For example, Minzoni et al. (2017) recorded around 50 km retreat of the Cosgrove Ice Shelf during the Holocene. Furthermore, ice loss post LGM was of an unbounded ice shelf on the outer ASE. It is ice shelf retreat when the ice shelf is bounded by topographic highs in the embayment, such as during the Holocene, that is important for understanding buttressing related to instabilities mentioned in the introduction (e.g., Pattyn, 2018).

Line 125: here, I think you need to highlight the distance between these sites (several hundred kilometres) and separate the sites into 2 or 3 records. The inner ASE (PS75/167 and PS75/160) and western outer ASE (VC436 and BC431) and eastern outer ASE (PS75/191 and PS75/190). The data in the plots should be separated into different panels accordingly and discussed separately.

Line 139: expand on how you discount carbonate ion concentration influence on you Mg/Ca ratios. As you mention in line 131, carbonate preservation is highly variable across the shelf suggesting carbonate ion concentration is highly variable.

Line 152: I am having trouble following this comparison. Maximum CDW temperatures of $>1.5^\circ\text{C}$ is confusing. Just put the range of CDW values.

Fig. 2: Separate the data into 4 panels i.e., red and blue circles in two panels and open squares, green triangles, and purple triangles in the other two. Change-point analysis should be conducted to see if there is a statistically significant shift in the data. Other tests are available such as ANOVA. There appears to be no change in the Mg/Ca ratios when split between inner and outer ASE, but statistical tests will be able to tell you.

Line 173: There is quite a lot of scatter in the Mg/Ca ratios. If you focus on the outer ASE records, I cannot see a clear cooling trend. Do you have any statistics that support this assertion? $\delta^{13}\text{C}$ is a bit clearer but there is still a lot of scatter.

Line 179: I think this quite a narrow view. There are lots of records of Holocene change, they are just not related to ocean forcing (e.g., Minzoni et al. 2017; Sproson et al., 2022; Johnson et al., 2014). Sea level was also a likely contributor to ice loss in the post LGM period that may not be as important now.

Line 197: This is misleading. I think you can say changes across the outer shelf but not embayment-wide as the inner ASE record starts at $\sim 10\text{ka}$.

Line 205: This sentence contradicts the next sentence. Lots of evidence for Holocene

retreat (e.g., Minzoni et al. 2017; Sproson et al., 2022; Johnson et al., 2014; 2017; 2020) and it is likely that the final 10s of kilometres of retreat (during the Holocene) are the most important for understanding contemporary ice loss as this is when ice shelves become pinned by plateaus causing the buttressing effect (Pattyn, 2018).

Line 213: CDW would have contributed to grounding line retreat. What about atmospheric and sea level forcing?

Line 218-221: This is a finding from Hillenbrand et al. (2017) and nothing new here except further support from Mg/Ca ratios.

Line 232-233: I think this is the main finding of the study and should be the main highlight. You can say a lot about incursion onto the outer shelf from 18 ka. Not a lot can be said about the inner ASE as that record begins at 10 Ka.

Line 244-255: How does this mechanism interact with sea-level rise? Could rising sealevel bring CDW further up the shelf? We see a rise from 20 to 8 ka BP in Fig. 3f.

Fig. 3a: the inner and outer records need to be separated out and treated independently.

Fig. 3c and d: Other records for SHW migrations are available that show continued poleward migration during the Holocene (Saunders et al., 2018; Voight et al., 2015; Ai et al. 2021). SHW migrations are a contentious issue and hotly debated so you should reflect this in the discussion as a possibility and not something that is unequivocally known.

Fig. 3e: More important here is to look at accumulation rates in relation to temperature for the WAIS Divide core (Fudge et al., 2016). Anomalous accumulation rates in the post-LGM era suggest atmospheric forcing of the Amundsen Sea sector of the WAIS.

Fig. 3g: sea-level is rising when CDW is present. How does sea-level influence CDW on the shelf? This is not discussed.

Line 301-305: This is pushing it to say the least. I don't think you can say this based on your data. Retreat continued into the Holocene; it is just ocean forcing played a diminished role (Sproson et al., 2022). Modelling suggests a greater contribution of tropical warming on warm air influencing the WAIS during the Holocene (Skinner et al., 2020; Dang et al., 2020; Scott et al., 2019).

Line 310: and surface melt from atmospheric warming (Ding, 2011; Steig et al., 2013; Wille et al., 2019).

Line 314: SHW poleward migration also caused warm, moist air to be delivered to West Antarctica leading to surface melting.

References:

- Ai, X. E. et al. Southern Ocean upwelling, Earth's obliquity, and glacial-interglacial atmospheric CO₂ change. *Science* 370, 1348–1352 (2020).
- Dang, H. et al. Pacific warm pool subsurface heat sequestration modulated Walker circulation and ENSO activity during the Holocene. *Sci. Adv.* 6, eabc0402 (2020).
- Ding, Q., Steig, E. J., Battisti, D. S. & Küttel, M. Winter warming in West Antarctica caused by central tropical Pacific warming. *Nat. Geosci.* 4, 398–403 (2011).
- Fudge, T. et al. Variable relationship between accumulation and temperature in West Antarctica for the past 31,000 years. *Geophys. Res. Lett.* 43, 3795–3803 (2016).
- Johnson, J. S. et al. Rapid thinning of Pine Island Glacier in the early Holocene. *Science* 343, 999–1001 (2014).
- Johnson, J. S. et al. The last glaciation of Bear Peninsula, central Amundsen Sea Embayment of Antarctica: Constraints on timing and duration revealed by in situ cosmogenic ¹⁴C and ¹⁰Be dating. *Quat. Sci. Rev.* 178, 77–88 (2017).
- Johnson, J. S. et al. Deglaciation of Pope Glacier implies widespread early Holocene ice sheet thinning in the Amundsen Sea sector of Antarctica. *Earth Planet. Sci. Lett.* 548, 116501 (2020).
- Minzoni, R. T. et al. Oceanographic influences on the stability of the Cosgrove Ice Shelf, Antarctica. *Holocene* 27, 1645–1658 (2017).
- Hillenbrand, C.-D. et al. West Antarctic Ice Sheet retreat driven by Holocene warm water incursions. *Nature* 547, 43–48 (2017).
- Pattyn, F. The paradigm shift in Antarctic ice sheet modelling. *Nat Commun* 9, 2728 (2018).

Saunders, K. M. et al. Holocene dynamics of the Southern Hemisphere westerly winds and possible links to CO₂ outgassing. *Nat. Geosci.* 11, 650–655 (2018).

Scott, R. C., Nicolas, J. P., Bromwich, D. H., Norris, J. R. & Lubin, D. Meteorological drivers and large-scale climate forcing of West Antarctic surface melt. *J. Clim.* 32, 665–684 (2019).

Skinner, C. B., Lora, J. M., Payne, A. E. & Poulsen, C. J. Atmospheric river changes shaped mid-latitude hydroclimate since the mid-Holocene. *Earth Planet. Sci. Lett.* 541, 116293 (2020).

Sproson, A. D., Yokoyama, Y., Miyairi, Y., Aze, T. & Totten, R. L. Holocene melting of the West Antarctic Ice Sheet driven by tropical Pacific warming. *Nat. Commun.* 13, 2434 (2022).

Steig, E. J. et al. Recent climate and ice-sheet changes in West Antarctica compared with the past 2,000 years. *Nat. Geosci.* 6, 372–375 (2013).

Voigt, I. et al. Holocene shifts of the southern westerlies across the South Atlantic. *Paleoceanography* 30, 39–51 (2015).

Wille, J. D. et al. West Antarctic surface melt triggered by atmospheric rivers. *Nat. Geosci.* 12, 911–916 (2019).

Reviewer #2

(Remarks to the Author)

Smith et al present new d¹³C and Mg/Ca data from the Amundsen Sea Embayment (ASE). The dataset advances our understanding of ocean forcing on ice sheet retreat. It extends the sparse temporal Holocene constraints published by Hillenbrand et al. (2017) *Nature*, back through to the last deglaciation for this critical West Antarctic glacier system. The power of the new data is the demonstration of two modes/states of the ocean system, and associated ice sheet (in)stability. During 18-10ka the average temperature of bottom water on the outer shelf was warmer compared to after 10ka when bottom water conditions were cooler. Temperature profiles near respective core sites show how incredibly warm CDW are, and proximal to the ice sheet.

The nature of continental shelf means that numerous core sites across the ASE are needed to measure both a meaningful number of rare foraminifera samples, and capture fully the last deglaciation. The novelty in the findings relies on the record generated in V436 which covers the deglaciation back to 18.2 ka. A step change in ocean conditions is observed which the author link to ice sheet retreat based on existing multi-proxy reconstructions of the region. The significance of this work is the clear shift in baseline bottom water temperature conditions, which puts into context the millennial variability observed since the early Holocene in Hillenbrand et al. (2017). This study suggests the same mechanism proposed by Hillenbrand et al. controlled the BWT on the shelf, with the additional details provided by ENSO observations on the controls on thermocline depth in setting the two modes of ocean forcing.

The data are of the highest quality analytically, the age models are clear, and the authors have dealt conservatively with extension of proxy data to temperature given the limits of the calibration/observations at the cooler end.

The manuscript is well written, comprehensively building on and advancing the latest understanding of this important sector of the ice sheet. I fully recommend publication of the manuscript. Below are considerations for the authors.

Comments:

Can the authors provide an explanation for the wide range on d¹³C, Mg/Ca. It is not only the late Holocene where a wide range of values are observed over ~1000-500 years, the deglacial section also show ~0.3-0.4 mmol/mol Mg/Ca range. Do the proxies capture the large seasonal variability in thermocline and water mass properties observed on the shelf? The strength of the new data presented is that it's time resolution is sufficient to observe the mode of forcing and provides context for centennial scale variability in CDW proxies. Can the author provide insight into controls on sea ice thresholds, or have any data to support threshold conditions in sea ice state? Very different sea ice conditions likely contribute to the depth of the mixed layer/thermocline.

More clarity around the temperature ranges is needed: seasonal variability in deep water temperature on Sup Fig 3, are these from mooring measurements over 1,2 or 3 years? How can the variability be larger than the min and max values at each site? It would be helpful to show where these profiles are from if multiple CTD stations were compiled? Or add additional plots to show the raw data from which the averages are compiled. These observations could support explanation of variability in proxy data related to previous comment?

Figure 2: Suggest you match the colour of squares in the d¹³C plot to the legend colour, as it's different to the colour of squares in the Mg/Ca plot beneath, and the colour of the square symbol in Figure 1.

Figure 3: misspelling of alkenones in c.

Reviewer #3

(Remarks to the Author)

The manuscript by Mawbey and co-authors deals with the retreat of the west Antarctic Ice Sheet and how it evolved during the last 20 ka. In detail the authors provide new geochemical constraints on the potential role of CDW in driving the retreat or the stabilization of WAIS.

Overall, this is an interesting and very timely study that provides an interesting dataset and interpretation. The conclusions drawn by the authors are for sure suitable for a journal like *Nature Communications*.

Saying this, I have to raise a couple of points and comments regarding the current version of the manuscript. Based on the

sum of these comments and suggestions and their implications I recommend to revise the submitted manuscript.

My points are listed below in the order they appear in the text.

Lines 59-60: by how much did the mentioned temperature increase in this interval? This would be good to mention to compare it with the findings of this study.

Data in figure 2: The authors mention that the Mg/Ca values and therefore ultimately their BWT data have been corrected for diagenetic coatings. They refer to the method section but there it is only stated that the correction was use following another study. Since the Mg/Ca data look very different between different sites and there is a large potential for trends being discussed as real shifts in temperate are simply changes between sites and therefore different preservation of the used foraminifera, it is necessary to show the Mn/Ca data and the responding corrections n a cross plot against Mg/Ca and do this site specific so the reader can judge how much trend and difference is site specific or how much might be a real signal. Without this information it is nearly impossible to check the reliability of the presented data.

Another aspect here would be the information of error bars for the data shown. How much of an error is added when the Mn/Ca correction is done? For the shown BWT data, this would add to the measurement error and the error of the used calibration. This combined error is nowhere in the manuscript shown or mentioned. Since the changes in Mg/Ca are rather small within a single core, large parts of the interpreted signal might be in error of the method used. I do not say (and also do not believe) that the signal is not true, but the reader needs to know this information to judge the reliability of the data presented.

Lines 189-191: the mentioned trend is only observed in a single datapoint that is a little bit higher than the others of site PS75/167 and for Site PS75/190 it is also only one point that is still relatively close to the values of the other (younger) data from this site. So this comes again back to the discussion about the error of the measurements.

Lines 218-221: What is this statement related to in figure 2? I do not see such a shift in the data (definitely not outside the estimated error).

Line 231: here the authors infer a 'pronounced decrease in BWT'. This is simply not seen in the data and needs much more discussion and careful phrasing when errors are involved.

Figure 3: The authors discuss and interpret the data of the last decades as a real increasing trend in BWT and infer that this is supporting the observations of a melting WAIS. However, the data are all very scattered for a single site and the observed trend is well in the range of the scatter observed for the other sites. So how can this be treated as a real trend?

Methods: It would be great to have some SEM images of the used foraminifera from the single sites to get a feeling for the preservation. Just under the light microscope it is impossible to judge recrystallization or dissolution.

Version 1:

Reviewer comments:

Reviewer #1

(Remarks to the Author)

The authors have adequately answered all of my points and this manuscript now represents a remarkable study of ocean drivers of Antarctic Ice Sheet loss worthy of publication in Nature Communications.

I have no further comments.

Reviewer #3

(Remarks to the Author)

The manuscript by Mawbey et al is a revised version of a manuscript I already reviewed during the initial submission. The authors have done a good job in answering most of the questions and comments the reviewers (including myself) raised. The revised manuscript is much better to understand and to follow and I think that the study and the underlying data are now better presented than before.

Saying this, however, I still have comments regarding the BWT reconstruction using benthic foraminiferal Mg/Ca the authors use to justify and interpret the inflow of colder waters onto the shelf region after approx. 10 ka. The authors have done a much better job to describe how the Mg/Ca data have been analyzed and how they were treated. However, looking at the now included errors of the data, all the trends and therefore the interpretation regarding warmer or colder water masses are well within the error of their analyzes (see new figure in SI). Especially in the final conclusion and establishment of the proposed interpretation, the authors repeatedly claim colder temperatures after 10 ka (speaking about 'pronounced decreases' in BWT and 'warmer BWT' before 10 ka), a fact that simply is not really seen in the data. Also the now used change-point analysis shows only trends and a change point if the used cores are combined. How does it look like when the

single cores are used? Is there really a solid and statistically sound change point? I would argue, that only if the data of multiple cores are combined, the change point and also the overall trend is visible. Combining cores, however, is prone to introduce different influences (preservation, geochemically slightly different water masses, etc.) a fact that is not convincingly described and discussed in the manuscript.

I have to say, that the data for C-isotopes are much more convincing than the Mg/Ca data, but with the former ones alone, the study misses a new and convincing dataset.

A second point I would like to raise (and would like to ask the authors to deal with in a revised version of the manuscript) is the assumption, that the water masses are evenly distributed across the shelf (i.e., inner to outer shelf are influenced by the same water mass), an assumption that is based on modern observations. How does this might change when you have completely different background conditions like an retreating marine ice shelf as it occurred during the study interval? Shouldn't this significantly change the outer vs. inner relationship and factors influencing it?

Version 2:

Reviewer comments:

Reviewer #3

(Remarks to the Author)

I would like to thank the authors for doing the requested changes in the manuscript. They have adequately answered all of my points (including the new error analysis).

I believe that this study will advance our knowledge of the AIS dynamics and recommend publication in Nature Communications.

Review for NCOMMS-25-17907-T

We thank the reviewers for their thorough and constructive feedback. Below we provide a point-by-point response to these comments (response in red). Line numbers refer to text edits in the 'clean' (rather than track-change) version.

Reviewer #1 (Remarks to the Author):

Mawbey et al. present carbon isotope and Mg/Ca data for the Amundsen Sea Embayment (ASE), a critical region for understanding the trajectory of future sea-level rise and relate this to the presence/absence of Circumpolar Deep Water (CDW), a probable driver of ice loss. The study extends records beyond the Holocene and is therefore an important contribution to understanding the long-term nature of the West Antarctic Ice Sheet. However, I find several problems that need to be addressed before publication:

1. D13C and Mg/Ca records from all sites are combined in Fig. 2 and 3a to represent changes in CDW versus AASW for the ASE in the text. However, Site PS75/167 and PS75/160 is in the inner ASE, several hundred kilometres from PS75/191, PS75/190, VC436 and BC431 on the outer shelf. These records should be separated and discussed separately: one record for the outer shelf discussing CDW upwelling; the other for the inner ASE discussing incursion further south (although this is already covered in Hillenbrand et al. 2017 but the Mg/Ca can be used to confirm these findings).

We thank the reviewer for their constructive comment regarding the structure of the results and the spatial distribution of our core sites. In response, we have restructured the Results section to present and describe the outer and inner shelf records separately (lines 181-204). We hope this distinction improves clarity and more explicitly acknowledges spatial variability across the shelf.

As noted in our original submission, the spatial and temporal variability in our dataset—split between outer shelf cores (PS75/192/190, VC436, and BC431) and inner shelf cores (PS75/160/167)—is an unavoidable consequence of the region's glacial history and the patchy preservation of microfossil-bearing sediments. These factors have constrained site (and sample) availability. It is worth reiterating that together with our US and German partners, we have recovered over ~400 sediment cores from the Amundsen Sea Embayment (ASE). Of these, the sites included in this study are the only ones suitable for detailed reconstruction of bottom-water temperature (BWT) and water mass properties over the deglacial to Holocene interval. While in an ideal scenario, sites such as VC436 would span the entire 18–0 kyr BP interval, and we would have comparable microfossil-rich cores from the mid-shelf for direct comparison with inner shelf sites, such records unfortunately do not exist at present.

Regarding the reviewer's question about the representativeness of the outer shelf sites for oceanographic conditions on the inner shelf: **we argue that the outer shelf records do provide meaningful constraints** on inner shelf ocean properties. Present-day oceanographic observations from around Antarctica suggest a broad coherence of water mass properties between the inner and outer shelf. Specifically, regardless of the distance between the ice sheet and the shelf break, the Antarctic Slope Front (e.g., see Whitworth III et al., 1985; Thompson et al., 2018), act as a relatively porous barrier to exchange between the Southern Ocean and the shelf seas (Schmidtko et al., 2014). On the shelf, near-seabed

ocean temperatures are typically either warm (CDW-like waters) or cold (AASW-like waters), with limited horizontal variability from the ice front to the shelf break.

To support this, we include a figure (Figure 1 below) from Zhou et al. (in prep) that presents a climatological synthesis of near-bottom temperatures across Antarctic shelf seas over the past 50 years. Two representative transects, extending from ice shelves to the open ocean in both warm shelf (most of West Antarctica) and cold shelf (most of East Antarctica) scenarios, illustrate the coherent nature of shelf water masses across spatial scales relevant to our study (see also Schmitdko et al. 2014).

In addition, numerical modelling supports the view that oceanographic signals recorded at a single deep location on the shelf can be representative of broader shelf conditions. Specifically, model simulations indicate that the residence time of shelf waters—that is, the time required to fully replace water properties through advection following a regime change—is relatively short. For the Amundsen Sea, residence times are on the order of a few years (Nakayama et al., 2014), while for larger Antarctic shelf seas, residence times are estimated to be on the order of a few decades (Naughten et al., 2021). This implies that, in the absence of persistent localized forcing, a coherent signal observed over centennial or longer timescales at a single site is likely to reflect conditions integrated across the broader shelf. Consequently, we consider the oceanographic signals recorded in our outer shelf cores to be representative of larger-scale changes across the ASE shelf, including areas closer to both present and former grounding lines.

Importantly, we observe remarkably consistent behaviour between the inner and outer shelf records during the critical transition around ~10 ka BP and extending through to the present day. This cross-shelf coherence supports the interpretation that the shelf was responding to the same large-scale oceanographic forcing. Furthermore, the spatial consistency of the signal along the outer shelf—persisting over thousands of years across a relatively small shelf width—suggests that the inner shelf remained in equilibrium with the outer shelf. This reinforces the validity of using outer shelf records as proxies for conditions influencing ice-sheet dynamics near the grounding line.

In summary, the reviewer has raised a valuable point, and we acknowledge that our original submission did not sufficiently explain the relationship (and linkages) between the inner and outer shelf sites. We have revised the manuscript to better articulate this connection and to justify the representativeness of the outer shelf records (lines 181-204 and 228-230).

Figure 1. Cold and warm oceanographic regimes on the Antarctic shelf (from Zhou et al., submitted) showing water mass consistency between outer and inner shelf settings. We also note that on palaeo timescales, the outer shelf signal is transmitted to the inner shelf instantaneously (years to decades). The data from Zhou et al. is made available under a

CCBY 4.0 license (<https://creativecommons.org/licenses/by/4.0/>) following the information in this link; <https://www.seanoe.org/data/00886/99787/>. No changes were made.

2. Following from the previous point, I think the records of outer shelf CDW presence/absence should be highlighted as the main finding of this study as these are new records that extend to ~18 ka BP. Any conclusions for the inner ASE should be toned down as these stem from the previous study of Hillenbrand et al. (2017) and start at ~10 ka BP.

We agree that the outer shelf core data are critical to our overall interpretation and have accordingly emphasized their importance in the revised manuscript (lines 181-204). However, we also wish to highlight that the inner shelf Mg/Ca data presented here are both novel and significant. While they follow the pattern previously inferred from $\delta^{13}\text{C}$ records, it is important to note that—aside from only few pilot Mg/Ca data published in Hillenbrand et al. (2017)—our study is the first to present a reconstruction of post-LGM Mg/Ca-based BWT from the Antarctic continental shelf.

Our earlier work (Hillenbrand et al., 2017) established that CDW was present on the inner shelf of the eastern ASE as soon as grounded ice had retreated from there, and that the transition from CDW- to AASW-dominated conditions occurred around 8 ka BP. However, due to the absence of direct proxy data for ocean temperature changes during the retreat phase before ca. 9 ka BP, we could not establish a robust causal link between ocean thermal forcing and grounded ice-sheet retreat from the outer to middle shelf. The additional new Mg/Ca data now provide that missing evidence, directly linking bottom-water warming and subsequent cooling to changes in ice-sheet dynamics.

Nevertheless, in line with the reviewer's recommendation, we have revised the manuscript to place greater emphasis on the outer shelf records, and we have added new text (lines 181-204) clarifying how oceanographic conditions on the outer and inner shelf relate to one another (lines 216-225).

3. The trends in the data need to be handled statistically instead of relying on the reader to pick out the perceived trends in the data, which are subjective. For example, change-point analysis could be used to determine if the jump at ~10 ka is statistically significant. I expect the $\delta^{13}\text{C}$ would see a significant jump, but the Mg/Ca ratios would not. This should be done after the inner ASE records (PS75/167 and PS75/160) are separated from the outer ASE (PS75/191, PS75/190, VC436 and BC431).

As requested, we have performed change-point (CP) analysis on both the Mg/Ca and the $\delta^{13}\text{C}$ data with corresponding text in the manuscript (lines 228-230), a new Supplementary Figure 6 and details of how this was done in the Methods (lines 573-584). We perform CP analyses on the inner/outer records as well as the composite record (for the reasons outlined above). All datasets yield consistent CPs.

Simple (segmented mean) analysis of the Mg/Ca data indicates significant CPs at 8248 yr BP (inner shelf) and 11043 and 10050 yr BP (outer shelf). The two CPs of the outer shelf Mg/Ca data CP indicate transitions to warmer (11043 to 10050 yr BP) and then cooler (<10050 yr BP) BWTs respectively. Analysis of the $\delta^{13}\text{C}$ data reveals CP at 7977 yr BP (inner shelf) and 10911 yr BP (outer shelf). Analysis of the composite datasets reveals broadly consistent results, yielding CPs at 10135 and 9113 yr BP (Mg/Ca) and 10911 and 7938 yr BP ($\delta^{13}\text{C}$), respectively. We attribute the difference in CPs between the Mg/Ca and $\delta^{13}\text{C}$ dataset to resolution, with the latter having more datapoints. When discussing the transition point between 'warm' deglacial and 'colder' Holocene conditions we now quote the time interval 10.1 to 7.9 ka BP. This is taken from the composite dataset 10135 to 7938 yr BP.

Note that the result of this analysis is broadly consistent with our original submission, which stated a change from high/low (low/high) at 10-9 ka BP for the Mg/Ca ($\delta^{13}\text{C}$) data.

Figure 2. Change-point analysis of Mg/Ca and $\delta^{13}\text{C}$ data using programme ‘changept’ in software package R. a. Inner-shelf Mg/Ca (CP at 8248 yr BP). b. Outer-shelf Mg/Ca with CPs at 11043 yr BP (warming) and 10050 yr BP (cooling). c. Inner shelf $\delta^{13}\text{C}$ (CP at 7977 yr BP). d. Outer-shelf $\delta^{13}\text{C}$ with a CP at 10911 yr BP. e-f. Composite Mg/Ca records showing CPs at 10135 and 9113 yr BP, and $\delta^{13}\text{C}$ CPs at 10911 and 7938 yr BP. Vertical dashed lines denote the CP, while horizontal black lines denote the segmented means.

4. You mention the carbonate ion effect but don't state how you determined if this effect was influencing your data. Some evidence to show constant carbonate ion concentrations or some other screening method would be vital here. The ASE is under influence from large amounts of freshwater discharge from nearby glaciers suggesting variable alkalinity across the shelf. Carbonate preservation also appears to be highly variable suggesting a dynamic carbonate ion concentration across the shelf.

We apologize if this was not clearly explained in our original submission. In response to the reviewer's comment, we have added clarifying text to the revised manuscript (lines 501-508).

To summarize: Previous studies have suggested that seawater carbonate ion concentration [CO_3^{2-}] could exert a secondary influence on foraminiferal Mg/Ca ratios—in addition to the

primary temperature control—particularly in *epifaunal* species, such as *Cibicidoides wuellerstorfi* and *C. mundulus*, which calcify at or just above the sediment–water interface (Elderfield et al., 2010; Elderfield et al., 2006). In contrast, *Trifarina angulosa*, the benthic foraminifera species used in our study, is an infaunal species, meaning it lives and calcifies within the top 1–5 cm of the sediment column (Mackensen et al., 1990; Schumacher, 2001). In this environment, porewaters tend to equilibrate rapidly with respect to carbonate chemistry, and the saturation state remains near zero. This minimizes the influence of ambient $[CO_3^{2-}]$ on calcification (Elderfield et al., 2006; 2010). Elderfield et al. (2010) demonstrated this effect for *Uvigerina* spp., and given both the close genetic relationship between *Uvigerina* and *Trifarina* spp. (Schweizer et al., 2005) and the similar Mg/Ca–BWT relationship between the two species (Mawbey et al., 2020), it is reasonable to assume that *Trifarina* responds similarly—i.e., that any $[CO_3^{2-}]$ effect on Mg/Ca is negligible.

For completeness, we conducted a simple sensitivity analysis. Elderfield et al. (2010) estimate a sensitivity of approximately 0.002 ± 0.001 mmol/mol per $\mu\text{mol/kg}$ change in $\Delta[CO_3^{2-}]$ for *Uvigerina* spp., with higher $[CO_3^{2-}]$ leading to elevated Mg/Ca values. Modern CDW typically has a $[CO_3^{2-}]$ concentration of $\sim 75\text{--}80$ $\mu\text{mol/kg}$, while Antarctic Surface Water (or Winter Water) reaches up to ~ 120 $\mu\text{mol/kg}$ (Bostock et al., 2011; Gottschalk et al., 2015; Yu et al., 2014). Assuming a conservative maximum $\Delta[CO_3^{2-}]$ of 40 $\mu\text{mol/kg}$ (a likely overestimate, given modification of CDW on the shelf), the resulting Mg/Ca bias would be approximately +0.08 mmol/mol (using *Uvigerina* sensitivity tests performed by Elderfield et al. 2010). This is small compared to the magnitude of Mg/Ca changes observed in our record. Moreover, since the main earliest Holocene transition in our data is characterized by a decrease in Mg/Ca values (interpreted as cooling and a reduced CDW influence), any $\Delta[CO_3^{2-}]$ effect would dampen the observed signal. Thus, the true temperature signal may be even stronger than inferred (see also discussion under *Trace metal analyses* in the ‘Methods’ section of Hillenbrand et al. 2017).

In summary, while we acknowledge that $\Delta[CO_3^{2-}]$ effects cannot be completely ruled out, the available evidence indicates that any bias is negligible and does not affect the core interpretations of our study. This is particularly true given that our conclusions are based on the relative magnitude of changes rather than on absolute temperature reconstructions. We therefore do not attempt to correct the record for $\Delta[CO_3^{2-}]$ nor do we use Mg/Ca–BWT calibrations that include carbonate ion corrections.

5. The wider interpretation is very selective and focuses on a marine geoscience perspective, a common problem among marine paleoclimate records, whilst neglecting to mention other drivers of ice loss since the LGM such as atmospheric and sea level forcing. More southerly SHW may drive CDW incursions onto the shelf, but it also brings warm, moist air to West Antarctica leading to surface melting. Sea-level rise from the LGM to Holocene would have destabilised ice sheets leading to dynamic calving. It is likely the combination of warm ocean and atmospheric currents and sea level rise would have combined to drive ice loss post LGM. This paper supports the ocean forcing contribution which is important but not the whole story. A wider context is therefore needed.

We appreciate the reviewer’s suggestion and have restructured the discussion to consider additional deglacial drivers beyond oceanic forcing. As noted in both our original submission and previous publications on the timing of ice sheet retreat in the ASE (e.g. Smith et al. 2011; 2014), global sea-level rise is likely to have contributed to grounding-line retreat during deglaciation. Given the relatively early deglaciation of the ASE shelf, however, atmospheric

and sea-level forcing probably only played a subordinate role in the post-LGM deglaciation when compared to ocean forcing (as we previously pointed out in Hillenbrand et al., 2017).

The exact contribution of sea-level rise cannot be directly assessed using the proxies in our study. While we acknowledge that rapid sea-level rise could have played a role—particularly during meltwater pulses—we are limited in our ability to quantify or disentangle this effect from the available data. We discuss this in the revised manuscript (lines 302-312, 322-325) and suggest that future fully coupled ocean-atmosphere-ice sheet modelling efforts are needed to better refine this interplay.

In contrast, we do not consider that atmospheric drivers, particularly changes in surface air temperature (SAT), have been the primary control, although a poleward shift in the SHW would, as the reviewer correctly states, not only drive (more) CDW onto the ASE shelf but also move warm air masses further south. West Antarctic ice core data indicate indeed that SATs increased during the outer to inner shelf deglaciation of the eastern ASE shelf from ca. 19 ka to 11 ka BP, when the modern temperature was finally reached (Cuffey et al., 2016; Fudge et al., 2016). In order to drive substantial surface melt and runoff sufficient to destabilize the ice sheet, however, SATs would likely have needed to be several degrees warmer than modern levels—but post-LGM SAT maxima in West Antarctica were less than 1.5 °C higher than present, and, importantly, these maxima were only reached after grounded ice had retreated from the ASE shelf (Cuffey et al. 2016; Fudge et al. 2016).

Consequently, while we now more explicitly acknowledge the role of atmospheric and sea-level forcing, we continue to interpret ocean-driven melting—linked to the incursion of warm CDW—as the dominant driver of grounding-line retreat in the ASE. However, we have also added additional text stating the atmospheric drivers are likely to have played a role in Holocene ice shelf and grounding line variations in some near-coastal parts of the embayment, e.g., Cosgrove Ice Shelf (lines 360-374).

6. We see a gradual increase in sea level and southward transition of the SHW in Fig. 3, but Mg/Ca ratios are flat. Why do we not see gradually more CDW on the shelf with this change? This suggests the $\delta^{13}\text{C}$ and Mg/Ca ratios are instead displaying a binary signal, i.e., CDW is either present or not, which needs to be discussed. Furthermore, other records for SHW migration show a continued poleward migration until the mid-Holocene. This may suggest ocean forcing is important before 10 ka but atmospheric or sea level forcing is important after 10 ka – an interesting finding that has not been discussed.

This is a good point and one we have also considered. We agree that CDW presence/absence is likely to be binary (for this proxy). In relation to the SHW, we suggest there might be a latitudinal threshold, beyond which CDW is forced on (or off) the shelf. We discuss this in our revision (lines 302-316, 387-399). However, it is also worth reiterating that previous work, including our own, shows that retreat (along Marie Byrd Land coast) since 10 ka has been very limited.

After these changes are made, I expect you will have an important study of outer shelf CDW changes that can inform on the contribution of ocean forcing on ASE ice loss, and maybe more muted CDW influence on continued loss during the Holocene. This finding should be placed within the context of other important forcing mechanisms such as atmospheric currents and sea-level rise.

Thank you, we appreciate your constructive review and agree that our findings are important.

Other comments:

Fig. 1 b and text referring to ASE reconstructions: this reconstruction and the relevant text refers to the pre-Holocene records and might be misleading. Several studies since have shown extensive ice shelf retreat during the Holocene. For example, Minzoni et al. (2017) recorded around 50 km retreat of the Cosgrove Ice Shelf during the Holocene. Furthermore, ice loss post LGM was of an unbounded ice shelf on the outer ASE. It is ice shelf retreat when the ice shelf is bounded by topographic highs in the embayment, such as during the Holocene, that is important for understanding buttressing related to instabilities mentioned in the introduction (e.g., Pattyn, 2018).

We thank the reviewer for highlighting an important oversight in our original submission, which relates to the distinction between changes to the WAIS grounding lines (which we define as those along the eastern Marie Byrd Land coastline – including Pine Island, Thwaites, Smith, Pope glaciers etc.) and changes to other glacier/ice shelves in the Amundsen Sea e.g., along the western Ellsworth Land coast. We have added additional text to more clearly define what part of WAIS we are referring to (lines 237-239). It was not our intention to overlook the important work relating to the Cosgrove Ice Shelf, which is discussed on lines 244-247 and 362-368.

Line 125: here, I think you need to highlight the distance between these sites (several hundred kilometres) and separate the sites into 2 or 3 records. The inner ASE (PS75/167 and PS75/160) and western outer ASE (VC436 and BC431) and eastern outer ASE (PS75/191 and PS75/190). The data in the plots should be separated into different panels accordingly and discussed separately.

As previously noted, we have followed the reviewer's recommendations and separated the outer/inner shelf data into different panels and discuss separately. We also refer to our earlier response re: outer shelf being indicative to inner shelf changes.

Line 139: expand on how you discount carbonate ion concentration influence on you Mg/Ca ratios. As you mention in line 131, carbonate preservation is highly variable across the shelf suggesting carbonate ion concentration is highly variable.

See corresponding response above.

Line 152: I am having trouble following this comparison. Maximum CDW temperatures of $>1.5^{\circ}\text{C}$ is confusing. Just put the range of CDW values.

We have modified the original text accordingly.

Fig. 2: Separate the data into 4 panels i.e., red and blue circles in two panels and open squares, green triangles, and purple triangles in the other two. Change-point analysis should be conducted to see if there is a statistically significant shift in the data. Other tests are available such as ANOVA. There appears to be no change in the Mg/Ca ratios when split between inner and outer ASE, but statistical tests will be able to tell you.

Done - see our earlier response.

Line 173: There is quite a lot of scatter in the Mg/Ca ratios. If you focus on the outer ASE records, I cannot see a clear cooling trend. Do you have any statistics that support this

assertion? D13c is a bit clearer but there is still a lot of scatter.

Yes, the Mg/Ca data are inherently noisy (see below) and of lower resolution compared to the $\delta^{13}\text{C}$ data. The clearest indication for a 'gradient' is seen in the $\delta^{13}\text{C}$ data. CP analysis indicates that this transition from low(high) to high(low) $\delta^{13}\text{C}$ (Mg/Ca) occurs between 10.3-7.8 ka BP.

Line 179: I think this quite a narrow view. There are lots of records of Holocene change, they are just not related to ocean forcing (e.g., Minzoni et al. 2017; Sproson et al., 2022; Johnson et al., 2014). Sea level was also a likely contributor to ice loss in the post LGM period that may not be as important now.

We recognise that the issue may stem from a lack of clarity in how we defined the West Antarctic Ice Sheet (WAIS) in the context of our study area. Specifically, when we refer to WAIS retreat to the inner shelf by ~10 ka BP, we are referring to the sector of the ice sheet along the eastern Marie Byrd Land coast, which includes Pine Island, Thwaites, Smith, and Kohler glaciers (see lines 237-238). This region likely contributed disproportionately to ice mass loss during deglaciation and is the primary focus of our reconstruction. It was not our intention to discount or neglect other records from the Amundsen Sea. Rather, our aim was to highlight the overall reduction in mass loss and the apparent grounding-line stability along the Marie Byrd Land sector during the Holocene, as constrained by our and other datasets. We agree with the reviewer that sea-level rise may have played a role in driving ice-sheet retreat (lines 306-312), and we stated this explicitly in both the original manuscript and relevant earlier work. However, we acknowledge that our current dataset does not allow us to isolate or directly assess the contribution of sea-level forcing, and we now clarify this further in the revised text (line 321-325). Moreover, we point out that along different parts of the western Marie Byrd Land coast (120-158 °W) grounded ice had retreated to the inner shelf already by 15.6 ka BP and 20.9 ka BP (Klages et al., 2014), respectively. This retreat well before Meltwater Pulse (MWP) 1A at 14.6 ka BP and the MWP at 19 ka, respectively, suggests that sea-level forcing played only a subordinate role in driving the deglaciation of the majority of the Amundsen Sea shelf.

Line 197: This is misleading. I think you can say changes across the outer shelf but not embayment-wide as the inner ASE record starts at ~10ka.

See our earlier response relating to Amundsen Sea wide inference. However, we have amended this sentence (lines 255).

Line 205: This sentence contradicts the next sentence. Lots of evidence for Holocene retreat (e.g., Minzoni et al. 2017; Sproson et al., 2022; Johnson et al., 2014; 2017; 2020) and it is likely that the final 10s of kilometres of retreat (during the Holocene) are the most important for understanding contemporary ice loss as this is when ice shelves become pinned by plateaus causing the buttressing effect (Pattyn, 2018).

We have re-structured this section, adding a discussion about Holocene glacier variability vs. WAIS retreat i.e., glaciers draining the Marie Byrd Land coast (lines 237-247 and 353-364). Furthermore, we point out that several of the papers cited in this regard by the reviewer (Johnson et al.) actually refer to ice-sheet thinning in the hinterland of the ASE but not grounding-line retreat. Both processes were apparently decoupled in the ASE, with the former lagging the latter by several kyrs. This is explained by the existence of large ice shelves that buttressed ice further upstream until the collapse of these ice shelves allowed

upstream thinning at ca. 10.6 ka BP in the Dotson-Getz Trough area (Smith et al. 2011; Johnson et al. 2020) and ca. 7.5 ka BP in the Pine Island Trough area (Hillenbrand et al. 2017; Johnson et al. 2021).

Line 213: CDW would have contributed to grounding line retreat. What about atmospheric and sea level forcing?

See earlier responses (and lines 306-316).

Line 218-221: This is a finding from Hillenbrand et al. (2017) and nothing new here except further support from Mg/Ca ratios.

Our previous work (Hillenbrand et al. 2017) indicated presence of warm water on the inner ASE shelf as soon as grounded ice had retreated from there, suggesting that this retreat was forced by CDW incursions. The lack of evidence about ocean properties during the earlier grounded ice retreat from the outer and middle shelf, however, prevented us from establishing a firm causal link between ocean forcing and ice-sheet retreat. We agree with the reviewer that our manuscript fills this gap by presenting observations going back ~20 ka BP, and thereby provide a new line of evidence that the retreat was indeed forced by the ocean.

Line 232-233: I think this is the main finding of the study and should be the main highlight. You can say a lot about incursion onto the outer shelf from 18 ka. Not a lot can be said about the inner ASE as that record begins at 10 Ka.

See our earlier response.

Line 244-255: How does this mechanism interact with sea-level rise? Could rising sea level bring CDW further up the shelf? We see a rise from 20 to 8 ka BP in Fig. 3f.

It's certainly possible but there is limited information on whether rising sea level facilitates enhanced delivery of the CDW onto the shelf. However, we do think that the combined impact of rising sea level and warming ocean was important in driving deglaciation.

Fig. 3a: the inner and outer records need to be separated out and treated independently.

Done.

Fig. 3c and d: Other records for SHW migrations are available that show continued poleward migration during the Holocene (Saunders et al., 2018; Voight et al., 2015; Ai et al. 2021). SHW migrations are a contentious issue and hotly debated so you should reflect this in the discussion as a possibility and not something that is unequivocally known.

We now acknowledge this debate (lines 387-395).

Fig. 3e: More important here is to look at accumulation rates in relation to temperature for the WAIS Divide core (Fudge et al., 2016). Anomalous accumulation rates in the post-LGM era suggest atmospheric forcing of the Amundsen Sea sector of the WAIS.

See our earlier response. We again refer to the temperature record from the WAIS Divide Ice Core since the LGM, which does not show SATs more than 1.5 °C higher than today (Cuffey

et al. 2016; Fudge et al. 2016). Poleward SHW shifts may, as the reviewer highlights, cause higher precipitation which may not be accompanied by an SAT increase (Fudge et al. 2016) or outweigh its effect by stabilising the ice sheet.

Fig. 3g: sea-level is rising when CDW is present. How does sea-level influence CDW on the shelf? This is not discussed.

See our earlier response.

Line 301-305: This is pushing it to say the least. I don't think you can say this based on your data. Retreat continued into the Holocene; it is just ocean forcing played a diminished role (Sproson et al., 2022). Modelling suggests a greater contribution of tropical warming on warm air influencing the WAIS during the Holocene (Skinner et al., 2020; Dang et al., 2020; Scott et al., 2019).

Again, we think this comes down to whether one refers to the main WAIS outlet systems along the southern ASE coast (Pine Island, Thwaites glaciers etc.) or small glacial outlets draining into bays along the eastern ASE coast. Our revised text should address this.

Line 310: and surface melt from atmospheric warming (Ding, 2011; Steig et al., 2013; Wille et al., 2019).

Yes, but further to our responses above, we consider our interpretation of incursions of CDW as the dominant driver of mass loss in the Amundsen Sea sector of the WAIS and 'the key variable in determining future sea-level rise from Antarctica' to be the most plausible. We discuss the role of atmospheric warming on lines 314-316 and 362-368.

Line 314: SHW poleward migration also caused warm, moist air to be delivered to West Antarctica leading to surface melting.

We agree, but presently the Amundsen Sea sector experiences only very limited surface melting which rarely occurs on more than 30 days per year (including years with extreme melt) and is almost exclusively restricted to the near-coastal areas along the eastern ASE coast (Donat-Magnin et al., 2020; Tedesco and Monaghan, 2009). Mass loss is dominated by calving and ice shelf melting and melting at the grounding lines by warm CDW. Peak Holocene temperatures less than 1.5°C higher than today are unlikely to have made a fundamental difference to this situation.

Reviewer #2 (Remarks to the Author):

Smith et al present new $\delta^{13}\text{C}$ and Mg/Ca data from the Amundsen Sea Embayment (ASE). The dataset advances our understanding of ocean forcing on ice sheet retreat. It extends the sparse temporal Holocene constraints published by Hillenbrand et al. (2017) Nature, back through to the last deglaciation for this critical West Antarctic glacier system. The power of the new data is the demonstration of two modes/states of the ocean system, and associated ice sheet (in)stability. During 18-10ka the average temperature of bottom water on the outer shelf was warmer compared to after 10ka when bottom water conditions were cooler. Temperature profiles near respective core sites show how incredibly warm CDW are, and proximal to the ice sheet.

The nature of continental shelf means that numerous core sites across the ASE are needed to measure both a meaningful number of rare foraminifera samples, and capture fully the last deglaciation. The novelty in the findings relies on the record generated in V436 which covers the deglaciation back to 18.2 ka. A step change in ocean conditions is observed which the author link to ice sheet retreat based on existing multi-proxy reconstructions of the region. The significance of this work is the clear shift in baseline bottom water temperature conditions, which puts into context the millennial variability observed since the early Holocene in Hillenbrand et al. (2017). This study suggests the same mechanism proposed by Hillenbrand et al. controlled the BWT on the shelf, with the additional details provided by ENSO observations on the controls on thermocline depth in setting the two modes of ocean forcing.

The data are of the highest quality analytically, the age models are clear, and the authors have dealt conservatively with extension of proxy data to temperature given the limits of the calibration/observations at the cooler end.

The manuscript is well written, comprehensively building on and advancing the latest understanding of this important sector of the ice sheet. I fully recommend publication of the manuscript. Below are considerations for the authors.

Comments:

Can the authors provide an explanation for the wide range on $\delta^{13}\text{C}$, Mg/Ca. It is not only the late Holocene where a wide range of values are observed over ~1000-500 years, the deglacial section also show ~0.3-0.4 mmol/mol Mg/Ca range.

Benthic foraminiferal Mg/Ca measurements are inherently noisy, even when analytical uncertainties are accounted for (de Nooijer et al., 2014; de Nooijer et al., 2009; Rosenthal et al., 1997; Sadekov et al., 2005)). This stochastic 'geological/biological' noise likely arises from a combination of factors, including vital effects, ontogenetic variability, and calcification processes, none of which are yet fully understood (e.g., Skinner and Elderfield, 2007). At least some of these factors are likely to be responsible for the $\delta^{13}\text{C}$ variability, too. Some of the observed variabilities in the Mg/Ca and $\delta^{13}\text{C}$ data may reflect genuine interannual or, much more likely, interdecadal (and in some instances possibly even intercentennial) changes in water mass properties (see response on resolution below), but given current limitations, we cannot confidently disentangle these signals from the underlying stochastic variability.

We also note that the magnitude and character of variability in our dataset are consistent with other benthic foraminiferal Mg/Ca records, including those produced in the 'Elderfield Lab' (e.g., Skinner et al. 2003; Skinner and Elderfield, 2007; Sadekov et al. 2014) as well as from a broad range of paleoceanographic studies from different settings and time periods (Lawson et al., 2024; Lear et al., 2015; Martin et al., 2002; Nürnberg and Groeneveld, 2006).

Do the proxies capture the large seasonal variability in thermocline and water mass properties observed on the shelf? The strength of the new data presented is that its time resolution is sufficient to observe the mode of forcing and provides context for centennial scale variability in CDW proxies. Can the author provide insight into controls on sea ice thresholds, or have any data to support threshold conditions in sea ice state? Very different sea ice conditions likely contribute to the depth of the mixed layer/thermocline.

This is a good question which we have considered carefully. We agree with the reviewer that a key strength of our new dataset is its temporal resolution, which is sufficient to investigate the mode of oceanic forcing and ice-sheet response. We also fully acknowledge that

thermocline depth—and by extension, BWT—is sensitive to a range of processes, including some seasonal ones, but especially interannual variability in sea ice, wind stress/wind curl, and surface heat fluxes (Dutrieux et al., 2014; Weber et al., 2014; Yang et al., 2024; Zheng et al., 2021). Supplementary Fig. 3 was intended to illustrate some of this variability, although we recognize that the figure caption may not have communicated this clearly (please see Figure 2 and our response to the related comment below).

That said, we do not believe our data can resolve seasonal-scale variability. The nature of benthic foraminiferal Mg/Ca records—combined with sedimentation rates and sample spacing—suggests that our data most likely reflect decadal to centennial-scale averages (see next response below). In this context, we interpret the signal as binary or stepwise (i.e., presence/absence or more/less CDW influence) rather than as capturing finer-scale seasonal changes. Seasonality has been proposed as a factor influencing foraminiferal growth windows in some environments (Fontanier et al., 2003; Skirbekk et al., 2016), and this has also been demonstrated based on $\delta^{13}\text{C}$ data for an epifaunal benthic foraminifera species (*Cibicides wuellerstorfi*) in the Southern Ocean (Mackensen et al., 1993). Seasonal habitat preferences of *Trifarina* spp., however, remain unstudied. For this reason and because of the temporal resolution of our datasets (see next response), we refrain from interpreting our data in terms of seasonal-scale variability in water mass properties. We assume, however, that the geochemical composition of the analysed foraminifera shells reflects mainly the oceanographic conditions during the spring-summer season, i.e., when plankton blooms occur and, therefore, *Trifarina* is most likely to feed and calcify. This is the same season when the CTD data were collected (see Figure 3 below).

Additionally, palaeo sea-ice reconstructions from the Amundsen Sea are scarce. We are aware of only two studies, i.e., Kim et al. (2021), which focuses on the past few centuries, and Lamping et al. (2020), who applied biomarker proxies (IPSO25 and modified PD-IPSO25) to a core spanning the Holocene. Importantly, however, both records reflect local polynya dynamics that are not representative for the wider ASE shelf as both cores were recovered from within the Amundsen Polynya (located on the inner shelf north of the Dotson and eastern Getz ice shelves). None of the cores studied by us lies within the Amundsen Polynya (see Fig. 1 in the main paper).

In the revised manuscript, we now explicitly state that our data are unlikely to resolve seasonal variability (line 203-205). We also note that ongoing work seeks to apply biomarker-based approaches (e.g., IPSO25, TEX86L) to the same cores presented here. These efforts may ultimately allow us to better link surface processes—such as sea-ice cover and surface heat fluxes (affecting sea-surface temperature [SST])—with bottom water properties in future studies.

More clarity around the temperature ranges is needed: seasonal variability in deep water temperature on Sup Fig 3, are these from mooring measurements over 1, 2 or 3 years? How can the variability be larger than the min and max values at each site? It would be helpful to show where these profiles are from if multiple CTD stations were compiled? Or add additional plots to show the raw data from which the averages are compiled. These observations could support explanation of variability in proxy data related to previous comment?

Thank you for highlighting this — we agree that the caption for Supplementary Figure 3 did not adequately describe the data plotted. We have now revised the caption to clarify that, for each site, we plot the average of all available CTD data within a 60 km radius around each core site spanning the full instrumental period (1994 to present), plus the maximum and minimum temperatures at the water depths of our records (see Figure 3, below). This approach is intended to illustrate the range of interannual variability in water column

properties at each location because the foraminifera have likely lifespans between ca. 1 and 13 years (Hayward et al., 2014; Hohenegger, 2018).

We acknowledge that temporal oceanographic variability on decadal timescales may account for some of the variability in our Mg/Ca data between samples of similar age from the same core. However, as noted above, the temporal resolution of our sediment core data is insufficient to isolate or resolve seasonal or even annual signals, and we assume the Mg/Ca value of each analysed shell to reflect an integrated multi-annual to decadal average in BWT because the lifespan of benthic foraminifera is usually 1-13 years (Hayward et al. 2014; Hohenegger 2018). As we combined ca. 30 foraminifera shells from each 1 cm thick sample horizon for one Mg/Ca measurement, the Mg/Ca data point is likely to give a decadal to centennial average in BWT given the wide range of sedimentation rates in our cores. This conclusion is based on the following considerations: One of the highest sedimentation rates in our cores is the early Holocene section (230-900 cm core depth) in core PS75/167, which was deposited between 8.3 and 10.4 ka BP (Hillenbrand et al. 2017), providing a sedimentation rate of 3.2 mm/yr. This implies that even if *Trifarina* specimens lived only in the top layer of the seabed sediments (0-1 cm) with an (unlikely) lifespan of just 1 year, a sample from the early Holocene section of this core is likely to provide a Mg/Ca signal averaged over 3 years. However, there is clear evidence that *Trifarina* lives down to a depth of 5 cm below the seafloor surface (Schumacher 2001), i.e., such a sample most likely also contains specimens that lived up to 15 years after the deposition of the sampled horizon, so the Mg/Ca value actually gives a signal integrated over 15 years. Conversely, one of the lowest sedimentation rates in our cores is the late Holocene section (133-230 cm core depth) in core PS75/160-1, which was deposited between 1.7 and 4.4 ka BP (Hillenbrand et al. 2017), providing a sedimentation rate of just 0.36 mm/yr. This implies a 1 cm thick sample from this section provides an Mg/Ca signal integrated over 28 years at best (*Trifarina* living only in the top centimeter) and over 140 years at worst (*Trifarina* living down to 5 cm within the seabed).

Figure 3. All available CTD data plotted as potential temperature (top panel) spanning the full instrumental period (1994 to present) within a 60 km radius of the core sites (bottom panel).

Figure 2: Suggest you match the colour of squares in the d13C plot to the legend colour, as it's different to the colour of squares in the Mg/Ca plot beneath, and the colour of the square symbol in Figure 1.

Changed, thanks.

Figure 3: misspelling of alkenones in c.

Changed, thanks.

Reviewer #3 (Remarks to the Author):

The manuscript by Mawbey and co-authors deals with the retreat of the west Antarctic Ice Sheet and how it evolved during the last 20 ka. In detail the authors provide new geochemical constraints on the potential role of CDW in driving the retreat or the stabilization of WAIS.

Overall, this is an interesting and very timely study that provides an interesting dataset and interpretation. The conclusions drawn by the authors are for sure suitable for a journal like Nature Communications.

Saying this, I have to raise a couple of points and comments regarding the current version of the manuscript. Based on the sum of these comments and suggestions and their implications I recommend to revise the submitted manuscript.

My points are listed below in the order they appear in the text.

Lines 59-60: by how much did the mentioned temperature increase in this interval? This would be good to mention to compare it with the findings of this study.

It is actually a relatively minor increase in temperature but significant in terms of melt potential; at 600–700 m in Pine Island Bay, seawater temperatures were ~ 0.2 °C higher in 2009 compared to 1994. Importantly this represents a 6% increase in the mean temperature above freezing ($T - T_f$), theoretically the key parameter scaling melt rates beneath the ice shelves. However, because we do not discuss absolute BWTs in our manuscript, we decided not to quote the temperature increase here.

Data in figure 2: The authors mention that the Mg/Ca values and therefore ultimately their BWT data have been corrected for diagenetic coatings. They refer to the method section but there it is only stated that the correction was use following another study. Since the Mg/Ca data look very different between different sites and there is a large potential for trends being discussed as real shifts in temperate are simply changes between sites and therefore different preservation of the used foraminifera, it is necessary to show the Mn/Ca data and the responding corrections n a cross plot against Mg/Ca and do this site specific so the reader can judge how much trend and difference is site specific or how much might be a real signal. Without this information it is nearly impossible to check the reliability of the presented data.

We added cross plots of Mg/Ca versus Mn/Ca (Supplementary Fig. 8 and shown below) and additional text (Methods; lines 531-543) to clarify how and why the Mn coating correction was applied to our Mg/Ca record. We acknowledge there is uncertainty regarding the Mn correction method, primarily because the Mg content within the Mn-coatings is not known. However, we note that the maximum decrease of Mg/Ca values due to possible Mg contribution from Mn-coatings is ~ 0.16 mmol/mol (Figure 5 below; Supplementary Fig. 7), with most of the decrease being below ~ 0.1 mmol/mol Mg/Ca, which falls well below our 2 s.e. (~ 0.27 mmol/mol) and is smaller than the magnitude of the earliest Holocene change of ~ 0.4 mmol/mol Mg/Ca. Therefore, this correction does not impact the interpretations presented in this study.

Figure 4. Mn/Ca vs Mg/Ca cross plots for (a) VC436; (b) PS75/167; (c) PS75/160, (d) PS75/190/192; (e) BC431.

Figure 5. Mg/Ca data (purple line) with adjustment for a potential diagenetic Mg contribution by assuming a magnesium/manganese (Mg/Mn) ratio of 0.15 ± 0.05 mol/mol in the diagenetic Mn-coating (yellow line) following Hillenbrand et al. (2017). This represents the 'Mn corrected Mg/Ca' data presented in our manuscript. Also shown are potential diagenetic Mg contributions assuming Mg/Mn ratios in diagenetic coatings of 0.20 and 0.25 mol/mol

following Hasenfratz et al. (2017). (a) PS75/190/192; (b) VC436; (c) PS75/160; (d) PS75/167; (e) all data. Error bar denotes 2 standard error (s.e.).

Another aspect here would be the information of error bars for the data shown. How much of an error is added when the Mn/Ca correction is done? For the shown BWT data, this would add to the measurement error and the error of the used calibration. This combined error is nowhere in the manuscript shown or mentioned. Since the changes in Mg/Ca are rather small within a single core, large parts of the interpreted signal might be in error of the method used. I do not say (and also do not believe) that the signal is not true, but the reader needs to know this information to judge the reliability of the data presented.

We thank the reviewer for pointing out the shortfalls of uncertainty analysis in our original submission. We have now added 2 sigma uncertainty (standard error) to the Mn-corrected Mg/Ca (and $\delta^{13}\text{C}$) data which covers the reconstructed parameter uncertainty and includes the analytical uncertainty, sampling uncertainty, uncertainty due to contamination as well as proxy uncertainty and stochastic noise (McPartland et al., 2024; Parnell et al., 2008).

More generally, given the calibration uncertainty for BWT, we refrained from propagating errors in our original submission. Calibration uncertainties are large and typically the absolute temperature values vary depending on the choice of Mg/Ca-temperature calibration. This is a particular problem for benthic Mg/Ca-temperature calibrations due to the difficulty of producing such calibrations (Mawbey et al. 2020), their increased uncertainty for low BWT ranges (due to the lack of data points in this range available for producing more constrained calibrations) and the inherent geological noise in benthic foraminiferal Mg/Ca records (see previous response). We note that our absolute BWTs are unrealistic (lines 155-160) and only present this BWT reconstruction as Supplementary Fig. 1.

We also reiterate that the magnitude of the earliest Holocene change (for the Mn corrected Mg/Ca data) is larger than the differences attached to the various Mn corrections (Figure 5 above, Supplementary Fig. 7).

Lines 189-191: the mentioned trend is only observed in a single datapoint that is a little bit higher than the others of site PS75/167 and for Site PS75/190 it is also only one point that is still relatively close to the values of the other (younger) data from this site. So, this comes again back to the discussion about the error of the measurements.

We have added 2 s.e. error bars to both the $\delta^{13}\text{C}$ and Mg/Ca data (Figure 2 in MS and Supplementary Fig. 7). However, our original text referred to the shift in Mg/Ca and $\delta^{13}\text{C}$ data around 10 ka BP (which is confirmed by CP analysis).

Lines 218-221: What is this statement related to in figure 2? I do not see such a shift in the data (definitely not outside the estimated error).

We have removed this statement – CP analysis does not identify a statistically significant change.

Line 231: here the authors infer a ‘pronounced decrease in BWT’. This is simply not seen in the data and needs much more discussion and careful phrasing when errors are involved.

CP analysis indicates a statistically significant change between 10.3-7.8 ka BP. However, in response to the reviewer’s comment we have reworded the text and also removed “pronounced”.

Figure 3: The authors discuss and interpret the data of the last decades as a real increasing trend in BWT and infer that this is supporting the observations of a melting WAIS. However, the data are all very scattered for a single site and the observed trend is well in the range of the scatter observed for the other sites. So how can this be treated as a real trend?

See earlier point – we have deleted this statement as the trend to warmer BWT is not seen in CP analysis.

Methods: It would be great to have some SEM images of the used foraminifera from the single sites to get a feeling for the preservation. Just under the light microscope it is impossible to judge recrystallization or dissolution.

Yes, dissolution and recrystallization are a particular issue in the Southern Ocean and this is why we carefully examined each foraminifer specimen under the light microscope prior to geochemical analyses. Only well-preserved specimens were used. Any discoloured, altered or broken specimens were excluded from further analysis. Although we acknowledge the limitations of using a light microscope to assess diagenetic alterations, it is typically not possible to check each foraminifer specimen under an SEM, especially given the number of samples analysed in this study with each sample containing ~30 specimens. However, we have now added several SEM and light microscope images of specimens used for geochemical analysis in this study (Supplementary Fig. 9 and Figure 6 below).

Figure 6. Representative SEM and light microscope images of benthic foraminifer *Trifarina angulosa* shells. a-c. BC431, 3 cm. d-e. PS75/190, 9 cm. f-g. PS75/160, 21cm. h. BC431, 1 cm.

References

- Bostock, H.C., Hayward, B.W., Neil, H.L., Currie, K.I., Dunbar, G.B., 2011. Deep-water carbonate concentrations in the southwest Pacific. *Deep Sea Research Part I: Oceanographic Research Papers* 58, 72-85.
- Cuffey, K.M., Clow, G.D., Steig, E.J., Buizert, C., Fudge, T.J., Koutnik, M., Waddington, E.D., Alley, R.B., Severinghaus, J.P., 2016. Deglacial temperature history of West Antarctica. *Proceedings of the National Academy of Sciences* 113, 14249-14254.
- de Nooijer, L.J., Hathorne, E.C., Reichart, G.J., Langer, G., Bijma, J., 2014. Variability in calcitic Mg/Ca and Sr/Ca ratios in clones of the benthic foraminifer *Ammonia tepida*. *Mar. Micropaleontol.* 107, 32-43.

de Nooijer, L.J., Langer, G., Nehrke, G., Bijma, J., 2009. Physiological controls on seawater uptake and calcification in the benthic foraminifer *Ammonia tepida*. *Biogeosciences* 6, 2669-2675.

Donat-Magnin, M., Jourdain, N.C., Gallée, H., Amory, C., Kittel, C., Fettweis, X., Wille, J.D., Favier, V., Drira, A., Agosta, C., 2020. Interannual variability of summer surface mass balance and surface melting in the Amundsen sector, West Antarctica. *The Cryosphere* 14, 229-249.

Dutrieux, P., De Rydt, J., Jenkins, A., Holland, P.R., Ha, H.K., Lee, S.H., Steig, E.J., Ding, Q., Abrahamsen, E.P., Schroeder, M., 2014. Strong Sensitivity of Pine Island Ice-Shelf Melting to Climatic Variability. *Science* 343, 174-178.

Elderfield, H., Greaves, M., Barker, S., Hall, I.R., Tripathi, A., Ferretti, P., Crowhurst, S., Booth, L., Daunt, C., 2010. A record of bottom water temperature and seawater $\delta O-18$ for the Southern Ocean over the past 440 kyr based on Mg/Ca of benthic foraminiferal *Uvigerina* spp. *Quat. Sci. Rev.* 29, 160-169.

Elderfield, H., Rickaby, R., Henderiks, J., 2006. How do marine carbonate Mg/Ca and Sr/Ca proxies constrain Cenozoic ocean history. *Geochim. Cosmochim. Acta* 70, A158-A158.

Fontanier, C., Jorissen, F.J., Chaillou, G., David, C., Anschutz, P., Lafon, V., 2003. Seasonal and interannual variability of benthic foraminiferal faunas at 550m depth in the Bay of Biscay. *Deep Sea Research Part I: Oceanographic Research Papers* 50, 457-494.

Fudge, T.J., Markle, B.R., Cuffey, K.M., Buizert, C., Taylor, K.C., Steig, E.J., Waddington, E.D., Conway, H., Koutnik, M., 2016. Variable relationship between accumulation and temperature in West Antarctica for the past 31,000 years. *Geophys. Res. Lett.* 43, 3795-3803.

Gottschalk, J., Skinner, L.C., Misra, S., Waelbroeck, C., Menviel, L., Timmermann, A., 2015. Abrupt changes in the southern extent of North Atlantic Deep Water during Dansgaard–Oeschger events. *Nature Geoscience* 8, 950-954.

Hayward, B.W., Figueira, B.O., Sabaa, A.T., Buzas, M.A., 2014. Multi-year life spans of high salt marsh agglutinated foraminifera from New Zealand. *Mar. Micropaleontol.* 109, 54-65.

Hillenbrand, C.D., Smith, J.A., Hodell, D.A., Greaves, M., Poole, C.R., Kender, S., Williams, M., Andersen, T.J., Jernas, P.E., Elderfield, H., Klages, J.P., Roberts, S.J., Gohl, K., Larter, R.D., Kuhn, G., 2017. West Antarctic Ice Sheet retreat driven by Holocene warm water incursions. *Nature* 547, 43-+.

Hohenegger, J., 2018. Foraminiferal growth and test development. *Earth-Sci. Rev.* 185, 140-162.

Johnson, J.S., Pollard, D., Whitehouse, P.L., Roberts, S.J., Rood, D.H., Schaefer, J.M., 2021. Comparing Glacial-Geological Evidence and Model Simulations of Ice Sheet Change since the Last Glacial Period in the Amundsen Sea Sector of Antarctica. *Journal of Geophysical Research: Earth Surface* 126, e2020JF005827.

Johnson, J.S., Roberts, S.J., Rood, D.H., Pollard, D., Schaefer, J.M., Whitehouse, P.L., Ireland, L.C., Lamp, J.L., Goehring, B.M., Rand, C., Smith, J.A., 2020. Deglaciation of Pope Glacier implies widespread early Holocene ice sheet thinning in the Amundsen Sea sector of Antarctica. *Earth Planet. Sci. Lett.* 548, 116501.

Kim, S.-Y., Lim, D., Rebolledo, L., Park, T., Esper, O., Muñoz, P., La, H.S., Kim, T.W., Lee, S., 2021. A 350-year multiproxy record of climate-driven environmental shifts in the Amundsen Sea Polynya, Antarctica. *Glob. Planet. Change* 205, 103589.

Klages, J.P., Kuhn, G., Hillenbrand, C.D., Graham, A.G.C., Smith, J.A., Larter, R.D., Gohl, K., Wacker, L., 2014. Retreat of the West Antarctic Ice Sheet from the western Amundsen Sea shelf at a pre- or early LGM stage. *Quat. Sci. Rev.* 91, 1-15.

Lamping, N., Müller, J., Esper, O., Hillenbrand, C.-D., Smith, J.A., Kuhn, G., 2020. Highly branched isoprenoids reveal onset of deglaciation followed by dynamic sea-ice conditions in the western Amundsen Sea, Antarctica. *Quat. Sci. Rev.* 228, 106103.

Larter, R.D., Anderson, J.B., Graham, A.G.C., Gohl, K., Hillenbrand, C.-D., Jakobsson, M., Johnson, J.S., Kuhn, G., Nitsche, F.O., Smith, J.A., Witus, A.E., Bentley, M.J., Dowdeswell, J.A., Ehrmann, W., Klages, J.P., Lindow, J., Cofaigh, C.Ó., Spiegel, C., 2014. Reconstruction of changes in the Amundsen Sea and Bellingshausen Sea sector of the West Antarctic Ice Sheet since the Last Glacial Maximum. *Quat. Sci. Rev.* 100, 55-86.

Lawson, V.J., Rosenthal, Y., Bova, S.C., Lambert, J., Linsley, B.K., Bu, K., Clementi, V.J., Elmore, A., McClymont, E.L., 2024. Controls on Sr/Ca, S/Ca, and Mg/Ca in Benthic Foraminifera: Implications for the Carbonate Chemistry of the Pacific Ocean Over the Last 350 ky. *Geochemistry, Geophysics, Geosystems* 25, e2024GC011508.

Lear, C.H., Coxall, H.K., Foster, G.L., Lunt, D.J., Mawbey, E.M., Rosenthal, Y., Sosdian, S.M., Thomas, E., Wilson, P.A., 2015. Neogene ice volume and ocean temperatures: Insights from infaunal foraminiferal Mg/Ca paleothermometry. *Paleoceanography* 30, 1437-1454.

Mackensen, A., Fuetterer, D.K., Grobe, H., Schmiedl, G., 1993. Benthic foraminiferal assemblages from the eastern South Atlantic Polar front region 35 and 75oS: Distribution, ecology and fossilization potential. *Mar. Micropaleontol.* 22, 33-69.

Mackensen, A., Grobe, H., Fuetterer, D.K., 1990. Benthic foraminiferal assemblages from the eastern Weddell Sea between 68 and 73S: Distribution, ecology and fossilization potential. *Mar. Micropaleontol.* 16, 241-283.

Martin, P.A., Lea, D.W., Rosenthal, Y., Shackleton, N.J., Sarnthein, M., Papenfuss, T., 2002. Quaternary deep sea temperature histories derived from benthic foraminiferal Mg/Ca. *Earth Planet. Sci. Lett.* 198, 193-209.

Mawbey, E.M., Hendry, K.R., Greaves, M.J., Hillenbrand, C.-D., Kuhn, G., Spencer-Jones, C.L., McClymont, E.L., Vadman, K.J., Shevenell, A.E., Jernas, P.E., Smith, J.A., 2020. Mg/Ca-Temperature Calibration of Polar Benthic foraminifera species for reconstruction of bottom water temperatures on the Antarctic shelf. *Geochim. Cosmochim. Acta* 283, 54-66.

McPartland, M.Y., Münch, T., Dolman, A.M., Hébert, R., Laepple, T., 2024. The Colors of Proxy Noise. *Clim. Past Discuss.* 2024, 1-20.

Nakayama, Y., Timmermann, R., Rodehacke, C.B., Schröder, M., Hellmer, H.H., 2014. Modeling the spreading of glacial meltwater from the Amundsen and Bellingshausen Seas. *Geophys. Res. Lett.* 41, 7942-7949.

Naughten, K.A., De Rydt, J., Rosier, S.H.R., Jenkins, A., Holland, P.R., Ridley, J.K., 2021. Two-timescale response of a large Antarctic ice shelf to climate change. *Nature Communications* 12, 1991.

- Nürnberg, D., Groeneveld, J., 2006. Pleistocene variability of the Subtropical Convergence at East Tasman Plateau: Evidence from planktonic foraminiferal Mg/Ca (ODP Site 1172A). *Geochemistry, Geophysics, Geosystems* 7.
- Parnell, A.C., Haslett, J., Allen, J.R.M., Buck, C.E., Huntley, B., 2008. A flexible approach to assessing synchronicity of past events using Bayesian reconstructions of sedimentation history. *Quat. Sci. Rev.* 27, 1872-1885.
- Rosenthal, Y., Boyle, E.A., Slowey, N., 1997. Temperature control on the incorporation of magnesium, strontium, fluorine, and cadmium into benthic foraminiferal shells from Little Bahama Bank: Prospects for thermocline paleoceanography. *Geochim. Cosmochim. Acta* 61, 3633-3643.
- Sadekov, A.Y., Eggins, S.M., De Deckker, P., 2005. Characterization of Mg/Ca distributions in planktonic foraminifera species by electron microprobe mapping. *Geochemistry, Geophysics, Geosystems* 6.
- Schmidtko, S., Heywood, K.J., Thompson, A.F., Aoki, S., 2014. Multidecadal warming of Antarctic waters. *Science* 346, 1227-1231.
- Schumacher, S., 2001. Mikrohabitatansprüche benthischer Foraminiferen in Sedimenten des Südatlantiks (Microhabitat preferences of benthic foraminifera in South Atlantic Ocean sediments), *Berichte zur Polar- und Meeresforschung = Reports on Polar and Marine Research*. Alfred-Wegener-Institut für Polar- und Meeresforschung, Bremerhaven, Germany, pp. 1-151.
- Schweizer, M., Pawlowski, J., Duijnste, I.A.P., Kouwenhoven, T.J., van der Zwaan, G.J., 2005. Molecular phylogeny of the foraminiferal genus *Uvigerina* based on ribosomal DNA sequences. *Mar. Micropaleontol.* 57, 51-67.
- Skinner, L.C., Elderfield, H., 2007. Rapid fluctuations in the deep North Atlantic heat budget during the last glacial period. *Paleoceanography* 22.
- Skinner, L.C., Shackleton, N.J., Elderfield, H., 2003. Millennial-scale variability of deep-water temperature and delta O-18(dw) indicating deep-water source variations in the Northeast Atlantic, 0-34 cal. ka BP. *Geochem. Geophys. Geosyst.* 4.
- Skirbekk, K., Hald, M., Marchitto, T.M., Junttila, J., Klitgaard Kristensen, D., Aagaard Sørensen, S., 2016. Benthic foraminiferal growth seasons implied from Mg/Ca-temperature correlations for three Arctic species. *Geochemistry, Geophysics, Geosystems* 17, 4684-4704.
- Smith, J.A., Hillenbrand, C.-D., Kuhn, G., Larter, R.D., Graham, A.G.C., Ehrmann, W., Moreton, S.G., Forwick, M., 2011. Deglacial history of the West Antarctic Ice Sheet in the western Amundsen Sea Embayment. *Quat. Sci. Rev.* 30, 488-505.
- Smith, J.A., Hillenbrand, C.D., Kuhn, G., Klages, J.P., Graham, A.G.C., Larter, R.D., Ehrmann, W., Moreton, S.G., Wiers, S., Frederichs, T., 2014. New constraints on the timing of West Antarctic Ice Sheet retreat in the eastern Amundsen Sea since the Last Glacial Maximum. *Glob. Planet. Change* 122, 224-237.
- Tedesco, M., Monaghan, A.J., 2009. An updated Antarctic melt record through 2009 and its linkages to high-latitude and tropical climate variability. *Geophys. Res. Lett.* 36.

- Thompson, A.F., Stewart, A.L., Spence, P., Heywood, K.J., 2018. The Antarctic Slope Current in a Changing Climate. *Rev. Geophys.* 56, 741-770.
- Weber, M.E., Clark, P.U., Kuhn, G., Timmermann, A., Sprenk, D., Gladstone, R., Zhang, X., Lohmann, G., Menviel, L., Chikamoto, M.O., Friedrich, T., Ohlwein, C., 2014. Millennial-scale variability in Antarctic ice-sheet discharge during the last deglaciation. *Nature* 510, 134-+.
- Whitworth III, T., Orsi, A.H., Kim, S.-J., Nowlin Jr., W.D., Locarnini, R.A., 1985. Water Masses and Mixing Near the Antarctic Slope Front, Ocean, Ice, and Atmosphere: Interactions at the Antarctic Continental Margin, pp. 1-27.
- Yang, H., Kim, T.-W., Kim, Y., Yoo, J., Park, J., Cho, Y.-K., 2024. Variability of Inflowing Current Into the Dotson Ice Shelf and Its Cause in the Amundsen Sea. *Geophys. Res. Lett.* 51, e2023GL105404.
- Yu, J., Anderson, R.F., Jin, Z., Menviel, L., Zhang, F., Ryerson, F.J., Rohling, E.J., 2014. Deep South Atlantic carbonate chemistry and increased interocean deep water exchange during last deglaciation. *Quat. Sci. Rev.* 90, 80-89.
- Zheng, Y., Heywood, K.J., Webber, B.G.M., Stevens, D.P., Biddle, L.C., Boehme, L., Loose, B., 2021. Winter seal-based observations reveal glacial meltwater surfacing in the southeastern Amundsen Sea. *Communications Earth & Environment* 2, 40.
- Zhou, S., Dutrieux, P., Giulivi, C. F., Meijers, A. J. S., Lee W. S., Kim, T.-W., Hattermann, T. and Janout, M. (submitted). The OCEAN ICE hydrography profiles compilation and climatology. *Earth System Science Data*

REVIEWER COMMENTS

Reviewer #1 (Remarks to the Author):

The authors have adequately answered all of my points, and this manuscript now represents a remarkable study of ocean drivers of Antarctic Ice Sheet loss worthy of publication in Nature Communications.

I have no further comments.

We thank Reviewer #1 for their thorough review and positive comments regarding our revision.

Reviewer #3 (Remarks to the Author):

The manuscript by Mawbey et al is a revised version of a manuscript I already reviewed during the initial submission. The authors have done a good job in answering most of the questions and comments the reviewers (including myself) raised. The revised manuscript is much better to understand and to follow and I think that the study and the underlying data are now better presented than before.

Thank you. We appreciate the efforts of the reviewer and are happy to provide further details.

Saying this, however, I still have comments regarding the BWT reconstruction using benthic foraminiferal Mg/Ca the authors use to justify and interpret the inflow of colder waters onto the shelf region after approx. 10 ka. The authors have done a much better job to describe how the Mg/Ca data have been analyzed and how they were treated. However, looking at the now included errors of the data, all the trends and therefore the interpretation regarding warmer or colder water masses are well within the error of their analyzes (see new figure in SI). Especially in the final conclusion and establishment of the proposed interpretation, the authors repeatedly claim colder temperatures after 10 ka (speaking about 'pronounced decreases' in BWT and 'warmer BWT' before 10 ka), a fact that simply is not really seen in the data.

We thank the reviewer for highlighting this point, as it prompted us to examine our error calculations more carefully.

An important first point, is that **our primary interpretations do not rely on absolute BWT values**, given the limitations of both global and regional Mg/Ca–temperature calibrations discussed in our original submission (and again below). For this reason—and as stated previously—we did not propagate analytical or calibration uncertainties in our initial submission.

In our revised submission we reported a two-standard deviation (2 SD) error calculated from the mean of the composite Mn corrected Mg/Ca dataset (± 0.2745 mmol/mol; $\pm 26.31\%$). **This was a mistake, for which we apologise. In retrospect we should have plotted the Mg/Ca data with the analytical uncertainty** as per the $\delta^{13}\text{C}$ data. Note that the analytical uncertainty of replicate measurements for Mg/Ca is very small (± 0.0057 mmol/mol ($\pm 0.55\%$)), so is typically not shown in Mg/Ca records e.g. Elderfield et al., 2010; 2012; Kender et al., 2016; Lawson et al., 2024; Yang et al., 2025). In contrast, one or two standard error or standard deviation is commonly applied when converting to temperature (see below), as this calculation encompasses multiple sources of error, particularly the calibration error.

Because the main focus of our manuscript is on the changes of Mg/Ca ratios (and not converted temperature values), and to be consistent with Mg/Ca literature - we quote analytical error in our 2nd revision. Note that in our earlier study (Hillenbrand et al., 2017), we quoted an uncertainty of ± 0.05 mmol/mol for Mg/Ca, representing the analytical uncertainty in the Mg/Mn composition of the diagenetic coating (e.g., 0.15 ± 0.05 mmol/mol).

We follow this approach but additionally incorporate the analytical uncertainty of the Mg/Ca measurement. Propagating both sources of analytical error yields a mean 2 SD uncertainty of ± 0.0503 mmol/mol. To further validate this, we have performed Monte Carlo simulations ($n = 10000$) to probabilistically evaluate uncertainties (e.g., Thornalley et al., 2018; Yu et al., 2019; Dai and Yu, 2025). Monte Carlo simulations produce a mean error of ± 0.0577 mmol/mol (Figure 1 below). These results are included in Supplementary Figure 1a and Table 1. Note that we also have deleted Supplementary Figure 1b since we do not refer to this in our interpretation.

Figure 1. 10000 Monte Carlo uncertainty simulations on Mn-corrected Mg/Ca dataset, resulting in a mean 2SD error of ± 0.0577 mmol/mol (see Supplementary Table 1).

We think that our revised error analysis is robust – and in line with Mg/Ca literature - but if the reviewer favours a different way of representing error, we are happy to follow their recommendations.

Ultimately there is a 0.62 mmol/mol shift in Mg/Ca data between the deglacial phase (18-10.1 ka) and the Holocene (7.9-0 ka). Given that our propagated analytical error is ± 0.0577 mmol/mol (Figure X) we have high confidence that the signal - and our interpretation - is robust. We also emphasize that the inferred transition from “warmer” to “cooler” conditions is consistent with the $\delta^{13}\text{C}$ data, which the reviewer describes below as “convincing”.

For completeness, we have also propagated uncertainties associated with the BWT calculations (Supplementary Figure 1) following standard methods and consistent with recent studies in *Nature Communications* (e.g., Yang et al., 2025 <https://www.nature.com/articles/s41467-025-62446-x>). This approach incorporates (i) the replicate analytical uncertainty (± 0.0057 mmol/mol), (ii) the Mn/Ca correction uncertainty (± 0.05 mmol/mol), and (iii) the calibration uncertainty (± 0.098 mmol/mol). The calibration uncertainty is estimated following Cl  roux et al. (2008), based on the standard error of the regression coefficient reported by Mawbey et al. (2020). Propagation of these uncertainties yields a temperature error of approximately 1.6  C for each sample.

We have updated Figures 2 and 3 in the main manuscript, as well as Supplementary Figures 1 and 7, to reflect these revised uncertainties. Details of the error analysis have been added to the Methods section, and all relevant figure captions have been updated accordingly. As noted above, we are happy to adopt an alternative uncertainty treatment if the reviewer has a preferred approach.

Also the now used change-point analysis shows only trends and a change point if the used cores are combined. How does it look like when the single cores are used? Is there really a solid and statistically sound change point? I would argue, that only if the data of multiple cores are combined, the change point and also the overall trend is visible. Combining cores, however, is prone to introduce different influences (preservation, geochemically slightly different water masses, etc.) a fact that is not convincingly described and discussed in the manuscript.

We politely point the reviewer to our previous rebuttal and supplementary information. We did perform CP analysis on both the composite dataset and 'single' cores. CPs are consistent for all composite/individual and Mg/Ca and $\delta^{13}\text{C}$ datasets.

I have to say, that the data for C-isotopes are much more convincing than the Mg/Ca data, but with the former ones alone, the study misses a new and convincing dataset.

Thank you - we agree that our $\delta^{13}\text{C}$ dataset is 'convincing' and indicates that warmer CDW-like waters prevailed on the Amundsen Sea shelf between 18–10 ka, while cooler, AASW-like waters dominated on the Amundsen Sea shelf after ~10–7.9 ka, while. As outlined in our original submission, the dominance of cooler waters after ~7.5 ka was established by our earlier work based on $\delta^{13}\text{C}$ and a limited number of Mg/Ca measurements (n=10; Hillenbrand et al., 2017, *Nature*, <https://doi.org/10.1038/nature22995>).

The strength of the present study lies in the integration of an extensive new Mg/Ca dataset (18–0 ka) with additional $\delta^{13}\text{C}$ data for the deglacial interval (18–10 ka). Together they provide a coherent and robust reconstruction of water-mass variability over the past 18 ka.

A second point I would like to raise (and would like to ask the authors to deal with in a revised version of the manuscript) is the assumption, that the water masses are evenly distributed across the shelf (i.e., inner to outer shelf are influenced by the same water mass), an assumption that is based on modern observations. How does this might change when you have completely different background conditions like an retreating marine ice shelf as it occurred during the study interval? Shouldn't this significantly change the outer vs. inner relationship and factors influencing it?

Yes, we are happy to provide additional details re: "how does this might change when you have completely different background conditions like a retreating marine ice shelf as it occurred during the study interval'..."

An important first point is that the ice sheet had already retreated to near its modern position before at least ~11 ka in the eastern part of our study area and before at least ~12–13 ka in its western part (Larter et al., 2014; see Fig. 1b), i.e. well before the observed Mg/Ca and $\delta^{13}\text{C}$ shifts. Therefore, the 'background conditions' have remained stable since that time, and there is no evidence to suggest that the pattern of shelf-sea circulation since this retreat and the present differed significantly from today. **A second point is that we have rapidly thinning ice shelves and rapidly retreating grounding zones in the Amundsen Sea today** and there is very little change in the CDW temperature as it traverses the shelf.

Thus, together with the modelling results highlighted in our earlier rebuttal (Nakayama et al., 2014; Naughten et al., 2021; Zhou et al., in prep.), which indicate residence time of shelf waters — that is, the time required to fully replace water properties through advection following a regime change — is relatively short and occurs over years to decades, we are confident that the water masses were evenly distributed across the shelf during both the deglacial phase and during the post-10 ka “modern” background conditions.

We have now added text to clarify this in the revised manuscript (line 227).

References

- Cléroux, C., Cortijo, E., Anand, P., Labeyrie, L., Bassinot, F., Caillon, N., Duplessy, J.-C., 2008. Mg/Ca and Sr/Ca ratios in planktonic foraminifera: Proxies for upper water column temperature reconstruction. *Paleoceanography* 23. <https://doi.org/10.1029/2007PA001505>
- Dai, Y., Yu, J., 2025. Contributions of biological and physical dynamics to deglacial CO₂ release from the polar Southern Ocean. *Nature Communications* 16, 2665. <https://doi.org/10.1038/s41467-025-57677-x>.
- Elderfield, H., Greaves, M., Barker, S., Hall, I.R., Tripathi, A., Ferretti, P., Crowhurst, S., Booth, L., Daunt, C., 2010. A record of bottom water temperature and seawater delta O-18 for the Southern Ocean over the past 440 kyr based on Mg/Ca of benthic foraminiferal *Uvigerina* spp. *Quat. Sci. Rev.* 29, 160-169. <https://doi.org/10.1016/j.quascirev.2009.07.013>.
- Elderfield, H., Ferretti, P., Greaves, M., Crowhurst, S., McCave, I.N., Hodell, D., Piotrowski, A.M., 2012. Evolution of Ocean Temperature and Ice Volume Through the Mid-Pleistocene Climate Transition. *Science* 337, 704-709. <https://doi.org/10.1126/science.1221294>
- Hillenbrand, C.D., Smith, J.A., Hodell, D.A., Greaves, M., Poole, C.R., Kender, S., Williams, M., Andersen, T.J., Jernas, P.E., Elderfield, H., Klages, J.P., Roberts, S.J., Gohl, K., Larter, R.D., Kuhn, G., 2017. West Antarctic Ice Sheet retreat driven by Holocene warm water incursions. *Nature* 547, 43-48. <https://doi.org/10.1038/nature22995>.
- Kender, S., McClymont, E.L., Elmore, A.C., Emanuele, D., Leng, M.J., Elderfield, H., 2016. Mid Pleistocene foraminiferal mass extinction coupled with phytoplankton evolution. *Nature Communications* 7, 11970. <https://doi.org/10.1038/ncomms1197>.
- Larter, R.D., Anderson, J.B., Graham, A.G.C., Gohl, K., Hillenbrand, C.-D., Jakobsson, M., Johnson, J.S., Kuhn, G., Nitsche, F.O., Smith, J.A., Witus, A.E., Bentley, M.J., Dowdeswell, J.A., Ehrmann, W., Klages, J.P., Lindow, J., Cofaigh, C.Ó., Spiegel, C., 2014. Reconstruction of changes in the Amundsen Sea and Bellingshausen Sea sector of the West Antarctic Ice Sheet since the Last Glacial Maximum. *Quat. Sci. Rev.* 100, 55-86. <http://dx.doi.org/10.1016/j.quascirev.2013.10.016>.
- Lawson, V.J., Rosenthal, Y., Bova, S.C., Lambert, J., Linsley, B.K., Bu, K., Clementi, V.J., Elmore, A., McClymont, E.L., 2024. Controls on Sr/Ca, S/Ca, and Mg/Ca in Benthic Foraminifera: Implications for the Carbonate Chemistry of the Pacific Ocean Over the Last 350 ky. *Geochemistry, Geophysics, Geosystems* 25, e2024GC011508. <https://doi.org/10.1029/2024GC011508>.
- Mawbey, E.M., Hendry, K.R., Greaves, M.J., Hillenbrand, C.-D., Kuhn, G., Spencer-Jones, C.L., McClymont, E.L., Vadman, K.J., Shevenell, A.E., Jernas, P.E., Smith, J.A., 2020. Mg/Ca-Temperature Calibration of Polar Benthic foraminifera species for reconstruction of bottom water temperatures on the Antarctic shelf. *Geochim. Cosmochim. Acta* 283, 54-66. <https://doi.org/10.1016/j.gca.2020.05.027>.

Nakayama, Y., Timmermann, R., Rodehacke, C.B., Schröder, M., Hellmer, H.H., 2014. Modeling the spreading of glacial meltwater from the Amundsen and Bellingshausen Seas. *Geophys. Res. Lett.* 41, 7942-7949. <https://doi.org/10.1002/2014GL061600>.

Naughten, K.A., De Rydt, J., Rosier, S.H.R., Jenkins, A., Holland, P.R., Ridley, J.K., 2021. Two-timescale response of a large Antarctic ice shelf to climate change. *Nature Communications* 12, 1991. <https://doi.org/10.1038/s41467-021-22259-0>.

Thornalley, D.J.R., Oppo, D.W., Ortega, P., Robson, J.I., Brierley, C.M., Davis, R., Hall, I.R., Moffa-Sanchez, P., Rose, N.L., Spooner, P.T., Yashayaev, I., Keigwin, L.D., 2018. Anomalously weak Labrador Sea convection and Atlantic overturning during the past 150 years. *Nature* 556, 227-230. <https://doi.org/10.1038/s41586-018-0007-4>.

Yang, Z., Lear, C.H., Barker, S., Elsey, J., Gasson, E., Rosenthal, Y., Slater, S.M., Thomas-Sparkes, A., 2025. Major sea level fall during the Pliocene M2 glaciation. *Nature Communications* 16, 7641. <https://www.nature.com/articles/s41467-025-62446-x>.

Yu, J., Menviel, L., Jin, Z.D., Thornalley, D.J.R., Foster, G.L., Rohling, E.J., McCave, I.N., McManus, J.F., Dai, Y., Ren, H., He, F., Zhang, F., Chen, P.J., Roberts, A.P., 2019. More efficient North Atlantic carbon pump during the Last Glacial Maximum. *Nature Communications* 10, 2170. <https://doi.org/10.1038/s41467-019-10028-z>.